# Transcriptional response to stress is pre-wired by promoter and enhancer architecture

Anniina Vihervaara[1,2], Dig Bijay Mahat[2], Michael J. Guertin[3], Tinyi Chu[4,5], Charles G. Danko[4], John T. Lis[2] & Lea Sistonen [1]

Programs of gene expression are executed by a battery of transcription factors that coordinate divergent transcription from a pair of tightly linked core initiation regions of promoters and enhancers. Here, to investigate how divergent transcription is reprogrammed upon stress, we measured nascent RNA synthesis at nucleotide-resolution, and profiled histone H4 acetylation in human cells. Our results globally show that the release of promoter-proximal paused RNA polymerase into elongation functions as a critical switch at which a gene's response to stress is determined. Highly transcribed and highly inducible genes display strong transcriptional directionality and selective assembly of general transcription factors on the core sense promoter. Heat-induced transcription at enhancers, instead, correlates with prior binding of cell-type, sequence-specific transcription factors. Activated Heat Shock Factor 1 (HSF1) binds to transcription-primed promoters and enhancers, and CTCF-occupied, non-transcribed chromatin. These results reveal chromatin architectural features that orient transcription at divergent regulatory elements and prime transcriptional responses genome-wide.

[1] Faculty of Science and Engineering, Cell Biology, Åbo Akademi University, Turku 20520, Finland. [2] Department of Molecular Biology and Genetics, Cornell University, Ithaca, New York 14853, USA. [3] Department of Biochemistry and Molecular Genetics, University of Virginia, Charlottesville, Virginia 22908, USA. [4] Department of Biomedical Sciences, The Baker Institute for Animal Health, Cornell University, Ithaca, New York 14853, USA. [5] Graduate Field of Computational Biology, Cornell University, Ithaca, New York 14853, USA. Correspondence and requests for materials should be addressed to J.T.L. (email: jtl10@cornell.edu) or to L.S. (email: lea.sistonen@abo.fi)

The plasticity of transcriptional programs is fundamental for all biological processes from cellular growth and differentiation to coordinated functions of tissues and organisms. The execution of distinct transcriptional steps has been extensively investigated at promoters of single genes, providing the basis for our current comprehension of the ordered interactions of DNA elements, transcriptional regulators, and transcription machinery[1]. Beyond the interactions at gene promoters, distal *cis*-acting regulatory elements have emerged as prominent determinants of cell type-specific transcription[2–7]. Our understanding of transcription is, however, severely hampered by the lack of information on how genes and distal regulatory elements are globally orchestrated in response to developmental or environmental signals, such as stress. Here, we identified the molecular features that determine transcriptional responses in the human genome upon exposure to an acute 30-min heat stress, which is known to rapidly reprogram RNA synthesis[8–10]. To profile the dynamic regulation of genome-wide transcription, we mapped the transcriptionally engaged RNA Polymerase (Pol) in human myeloid/erythroid leukemia K562 cells prior to and upon heat shock using Precision nuclear Run-On sequencing (PRO-seq)[11]. Importantly, global run-on techniques enable strand-specific, nucleotide-resolution measures of nascent RNA synthesis, empowering the accurate mapping of gene expression[11–18] and identification of RNA-producing enhancers across the whole genome with high sensitivity[19, 20].

Heat shock factor 1 (HSF1) is the prime regulator of heat stress-induced transcription, and a key player for maintaining protein homeostasis in eukaryotic cells and organisms[21]. Upon activation, HSF1 rapidly binds to Heat Shock Elements (HSEs) at hundreds of genomic loci, both at genes and intergenic regions, as has been shown in yeast[22], round worm[23], fly[24, 25], mouse[16, 26] and human cells[27, 28]. Despite the central role of HSF1 in coordinating stress-induced transcription, several classes of genes that are activated by stress stimuli are not bound by HSF1[16, 22, 27]. Moreover, HSF1 binds to a number of genes that do not show induction upon stress[16, 22, 24, 25, 27, 28]. HSF1-driven transcription depends on the cell type, cell cycle phase, and is integrated into the metabolic and pathological state of the organism[21, 26–36]. Consistent with the condition-dependent transcriptional reprogramming, the chromatin that is targeted by HSF1 has been shown to reside generally in an open conformation prior to HSF1 activation[25, 28], which suggests that the local chromatin environment pre-conditions transcriptional responses. However, the precise chromatin configuration that determines the access of HSF1, or any other transcription factor, to DNA and the exact mechanistic control that underlies the transcriptional outcome at genes and distal regulatory elements have remained elusive.

In this study, we investigate the mechanisms that coordinate transcription of genes and enhancers in the human genome. By rapidly provoking a transcriptional change by heat shock and directly measuring RNA synthesis from genes and distal regulatory elements, our results provide a global view on transcriptional reprogramming. Nucleotide-resolution maps of nascent transcription complexes, in the context of the local chromatin environment, reveal how transcriptional responses are architecturally pre-set and rapidly adjusted in human cells. Moreover, deciphering of gene networks and their distal regulatory elements uncovers functional mechanisms that induce hundreds and repress thousands of genes, simultaneously establishing a distinct enhancer repertoire and chromatin state in stressed human cells. In particular, our results demonstrate how inhibition of the pause–release of promoter–proximal Pol II clears transcription complexes from the majority of transcribed genes upon stress, and how the consequently increased free Pol II in the cell is directed to the coding strands of the activated genes, as well as to enhancers that are primed by lineage-specific transcription factors.

## Results

**Rapid transcriptional reprogramming upon stress.** To investigate transcriptional programs in high-resolution, we performed PRO-seq and quantified nascent RNA synthesis in human erythroleukemia K562 cells under optimal growth conditions (NHS) and upon a 30-minute heat shock at 42 °C (HS). Since PRO-seq detects a single de novo-added nucleotide at the active site of each nascent transcript, it provides the locations of transcriptionally competent Pol complexes at a single-nucleotide resolution. PRO-seq does not discern which Pol is producing the RNA, but because our present study focuses on genes and enhancers, which are predominantly transcribed by Pol II[3, 12, 19, 37–39], we refer to Pol II as the source of the PRO-seq signal analyzed hereon. By utilizing the strand specificity of PRO-seq, the Pol II density was selectively quantified on the coding strand of each gene, scoring the promoter–proximal pause site as a 50-nucleotide (nt) window from a region spanning −100 nt to +400 nt from the annotated transcription start site (TSS), and measuring transcription of the gene body from +500 nt from the TSS to −500 nt from the polyA site (Fig. 1a, Supplementary Data 1). We analyzed upstream-divergent transcription, which is a common feature of mammalian promoters[12, 19, 20, 40–42], by mapping the position and intensity of Pol II at the anti-sense strand, upstream of each gene (Fig. 1a). The PRO-seq experiments were prepared and sequenced in two biologically-independent replicates, which strongly correlated (rho > 0.97; Supplementary Fig. 1a), and accurate comparison of transcriptional programs in NHS versus HS was ensured by normalizing each data set against the 3′-regions (+100 kb from TSS to −0.5 kb from polyA site) of long (>150 kb) genes, where the advancing or receding wave of transcription had not proceeded during the 30-min HS treatment (Supplementary Fig. 1b).

Identification of differentially transcribed genes in NHS versus HS revealed 778 significantly upregulated and 6122 significantly downregulated genes upon acute stress (Fig. 1b and Supplementary Data 1). Beyond the large number of heat-responsive genes, the profound transcriptional reprogramming upon acute stress was evident by the prompt changes at individual genes, as exemplified by the heat-induced autophagocytosis mediator *BAG3* (Fig. 1a), heat shock protein *HSPH1* (also known as *HSP110*; Supplementary Fig. 1c), as well as by the repressed eukaryote elongation factor *EEF1A2* (Supplementary Fig. 1d).

**Reprogramming of genes is defined at Pol II pause–release.** Transcription is primarily regulated at the steps of Pol II recruitment to promoters and subsequent promoter–proximal pause–release, which prompted us to determine whether these steps coordinated transcriptional reprogramming in heat-stressed cells. Upon acute stress, the average signal intensity of Pol II increased at the promoter–proximal pause site of all actively transcribed genes (Fig. 1c). This striking gain in the Pol II density near the TSS demonstrated that Pol II recruitment was not the rate-limiting step in stressed cells. Instead, the rapid and global stress-induced halt on gene expression was enforced by inhibiting the release of Pol II into productive elongation, a phenomenon that occurred on virtually every downregulated gene (Fig. 1d). Inhibiting the release of Pol II caused a robust increase in the pausing index (Fig. 1e and Supplementary Fig. 1e), a tightening of the pause site towards the TSS (Supplementary Fig. 1f), and a

receding transcriptional wave that cleared the gene body from transcribing Pol II (Supplementary Fig. 1b).

Upregulated genes, on the contrary, increased the rate of initiation and the release of Pol II to productive elongation as evidenced by the higher Pol II density at the pause site and along the gene body (Fig. 1c,d). Importantly, the upregulated genes maintained the proportion of Pol II at the promoter–proximal region versus the gene body (Fig. 1e and Supplementary Fig. 1e),

which indicated that upregulated genes effectively coupled the release of Pol II into the productive elongation with the rapid filling of the freed pause site, enabling immediate initiation of a new round of transcript production.

Taken together with recent studies in *Drosophila*[18] and mouse[16], our mapping of RNA synthesis in human cells identified the step of Pol II pause–release as the essential switch at which the gene's response to stress was defined. A fundamental feature

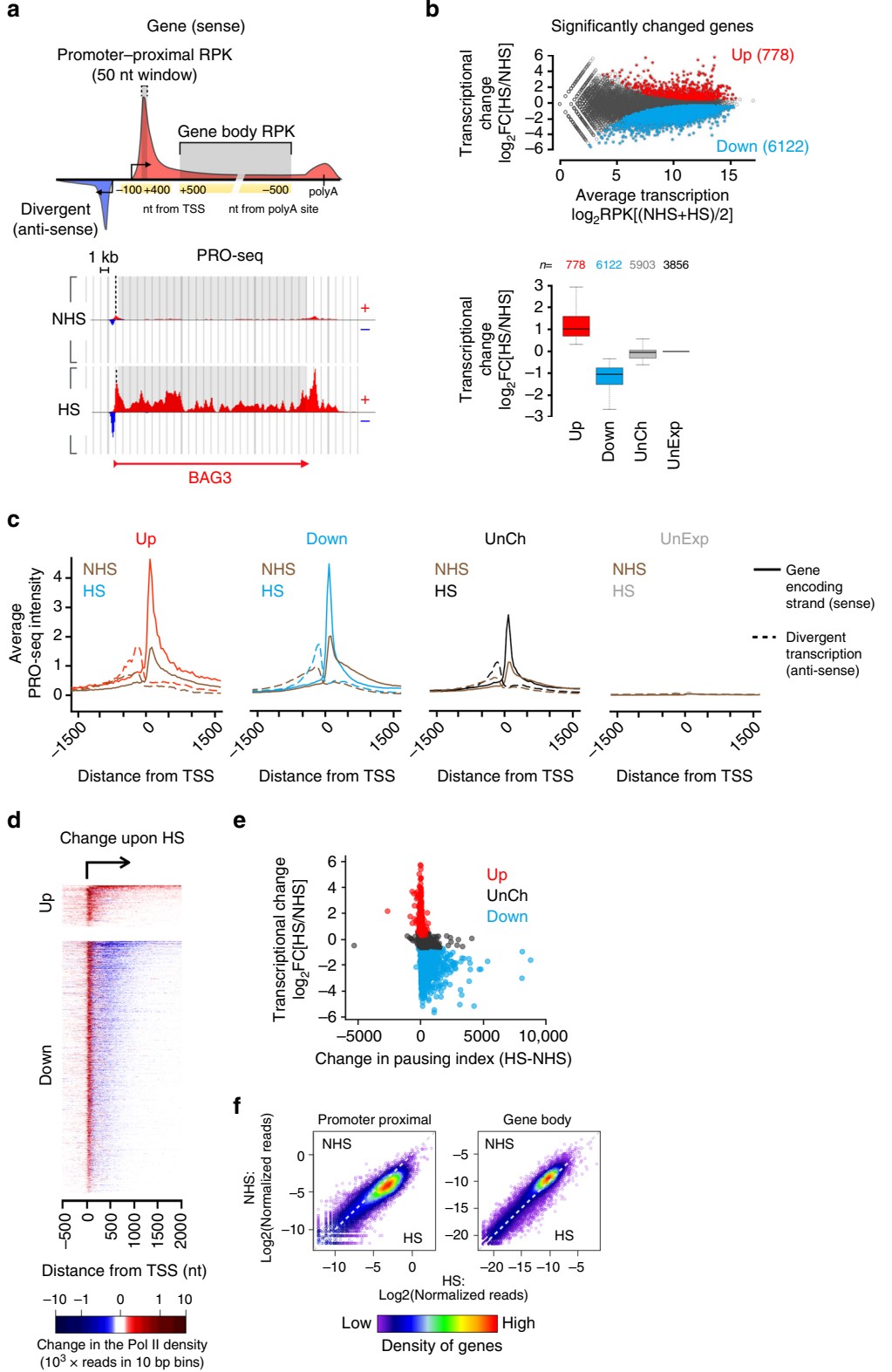

of transcriptional regulation upon stress was that the Pol II molecules that were productively elongating when the stress occurred then continued to the end of the gene. Therefore, the Pol II molecules that were released from the ends of thousands of genes provided a source of free Pol II, which could be exploited for an instant filling of the unoccupied pause sites by mass action. This concept was supported by the substantial change in the genomic localization of Pol II upon stress, diminishing from the gene bodies and concurrently accumulating at the promoter–proximal pause sites (Fig. 1f). Our results offer a simple explanation for how regulation of the single step of promoter–proximal pause-release could provide a switch at which an immediate and global transcriptional response is determined.

**Emergence of a distinct repertoire of regulatory elements**. The human genome encodes a large variety of RNA species, including the unstable divergent transcripts (eRNAs) that arise from active enhancers[3, 6, 19]. Genome-wide enhancer activity was previously assessed by measuring broad histone marks or by identifying transcription initiation sites of 5′-capped RNAs[19, 43–48]. We took advantage of the sensitivity of PRO-seq for analyzing both stable and unstable classes of RNA to: (1) identify the repertoire of RNA-producing distal regulatory elements in NHS and HS conditions and (2) quantify their transcriptional activity at high spatial and temporal resolution (Fig. 2 and Supplementary Data 2). We identified the precise locations of active Transcription Regulatory Elements (TREs) using a discriminative Regulatory Element detection algorithm (dREG)[20] and refined the regions between peaks of divergent Pol II initiation using dREG-HD (Methods section). Next, TREs were classified into promoters and distal TREs (dTREs), a class of regulatory elements that includes RNA-producing enhancers. This classification is based on the assumption that at least one of the divergent initiation sites originating from promoters will produce a stable mRNA (Fig. 1a), whereas dTREs produce unstable transcripts in both directions[19] (Fig. 2a). The coordinates of dTREs largely overlap with enhancers that actively produce eRNAs. However, since there is no functional evidence that every dTRE has enhancer activity, and a dTRE could also negatively influence transcription[49], we define the dTREs to include transcribed enhancers as well as other distal regulatory elements that produce unstable divergent transcripts.

Our novel computational tool, dREG-HD, identified from the PRO-seq data 16,723 dTREs in NHS, and 21,768 dTREs upon HS, of which 7764 occurred in both conditions (Fig. 2a, Supplementary Data 2). Hence, these analyses revealed the clearly distinct set of distal regulatory elements that emerged during 30 min of heat stress in human cells. Quantifying the

Pol II density along the length of each identified dTRE in a strand-specific manner (Fig. 2a), demonstrated that 5010 dTREs were significantly upregulated and 1628 dTREs significantly downregulated upon stress (Fig. 2b and Supplementary Fig. 2a). The immediate emergence of the stress-specific repertoire of dTREs is illustrated by heatmap analyses that depict the Pol II density across individual regulatory elements prior to and upon heat shock (Fig. 2c). Beyond detecting significantly changed dTREs by the total PRO-seq read count, a number of dTREs displayed a comparable presence but remarkably changed distribution of the engaged Pol II complexes. This changed profile of nascent RNA synthesis could be due to accumulation of the paused Pol II or altered usage of dTREs in a regulatory element cluster (examples shown in Supplementary Fig. 2b), further expanding the repertoire of heat-responsive regulatory elements. Overall, our sensitive and high-resolution mapping of dTREs indicated that distal regulatory elements were rapidly reprogrammed in stressed cells. The prompt change in the enhancer landscape, including the greater number of detected dTREs, raises an intriguing possibility that the free pool of Pol II that became available from the downregulated genes could facilitate also the immediate tuning-up of the enhancer repertoire.

**Histone acetylation at TREs increases with Pol II density**. Transcription is a dynamic process where the chromatin state and the Pol II machinery influence each other[50]. To investigate the simultaneous coordination of transcription and chromatin, we assessed chromatin state by measuring acetylation of histone H4 with an antibody that recognizes H4 acetylation at lysines 5, 8, 12, and 16. Consistent with previous studies[51, 52], chromatin immunoprecipitation coupled to deep sequencing (ChIP-seq) identified prominent acetylation of histone H4 at the promoters of transcribed genes. As shown in Supplementary Fig. 3a, histone H4 acetylation at the gene promoters was progressively higher concomitantly with the genes' transcriptional activity. In comparison, histone H4 acetylation could not be detected at the untranscribed genes, either at the promoter region or across the gene body (Supplementary Fig. 3a). The histone H4 acetylation, furthermore, positively correlated with chromatin accessibility, measured as DNAseI hypersensitivity, both at promoters and across the genome (Supplementary Fig. 3a).

Upon stress, the pattern of histone H4 acetylation underwent a drastic change both at genes (Fig. 3a–c) and dTREs (Fig. 3d, e, and Supplementary Fig. 3b,c). At transcribed genes, the promoters showed hyper-acetylation of histone H4 regardless of whether the Pol II was entering into productive elongation (upregulated genes), or it was predominantly paused and not

**Fig. 1** Rapid transcriptional reprogramming of genes is defined at the step of promoter-proximal pause-release. **a** Schematic representation (upper panel) of promoter-proximal region (−100 to +400 nt from TSS) from which the Pol II pause site was scored as a 50-nt window, and gene body (+500 nt from TSS to −500 nt from polyA site) from which the average Pol II density across the coding region was measured. Divergent transcription is depicted upstream of the gene, and the transcription initiation sites towards the sense and anti-sense are indicated with arrows. The lower panel illustrates the strand-specific scanning of Pol II density at promoter–proximal (*dashed line*) and gene body (*grey box*) regions at *BAG3* gene prior to (NHS) and upon (HS) heat shock. **b** MA-plot (upper panel) showing the heat-induced transcriptional change at the coding regions of individual genes. Genes with significantly upregulated (Up) or downregulated (Down) transcription upon HS are colored red and blue, respectively. The lower panel indicates the number and transcriptional change of genes that were significantly upregulated, downregulated or remained unchanged (UnCh) upon acute heat stress, or that were not transcribed (UnEp) prior to or upon HS in human K562 cells. **c** Strand-specific average intensity of transcriptionally engaged Pol II at the TSS of upregulated (Up), downregulated (Down), unchanged (UnCh) and unexpressed (UnEp) genes. Coding strand is indicated with solid, divergent strand with dashed line. **d** Heatmap depicting the change in the Pol II density at the coding strand of significantly changed genes upon acute stress. **e** The change in the pausing index at individual upregulated, downregulated and unchanged genes. **f** Comparison of PRO-seq reads prior to (NHS) and upon heat shock (HS) at promoter-proximal and gene body regions of each gene. The density of genes in the scatter plot is indicated with the color scale

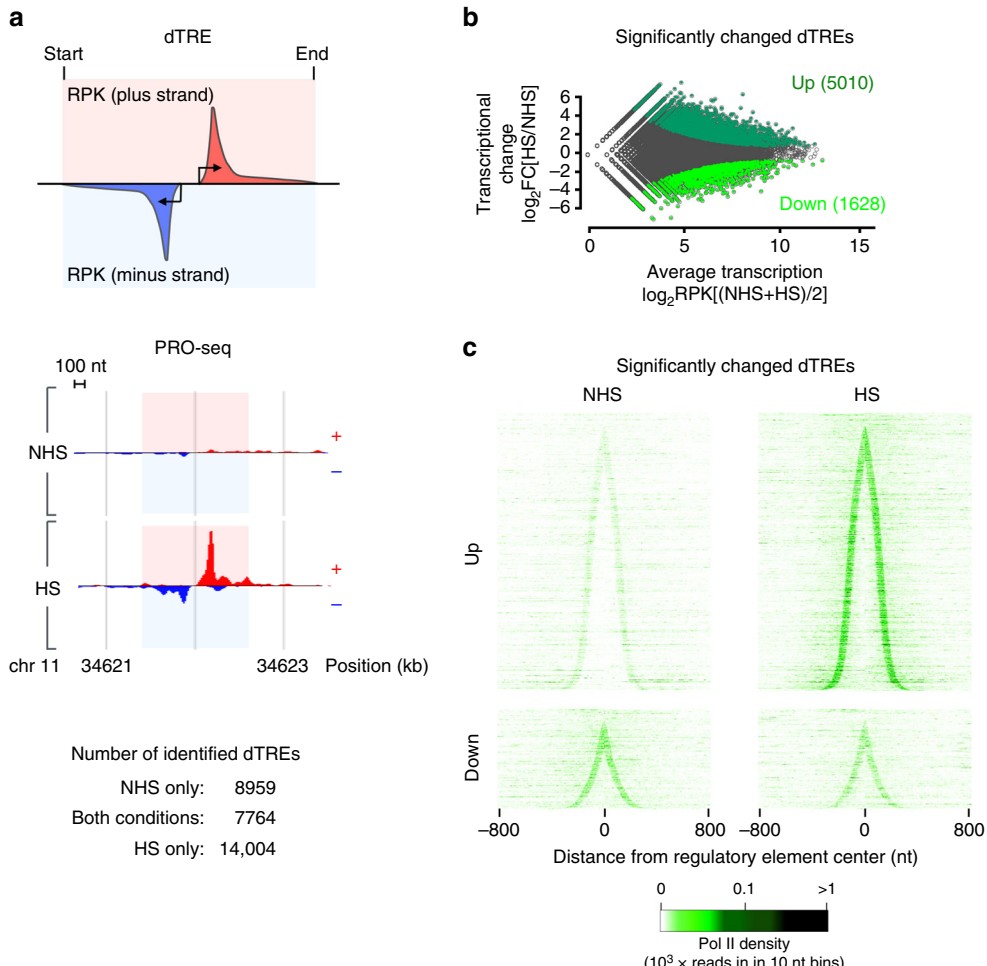

**Fig. 2** Immediate emergence of a stress-specific repertoire of active regulatory elements. **a** Schematic representation (*upper panel*) and a browser shot example (*lower panel*) of a transcribed distal regulatory element (dTRE) showing the characteristic pattern of short divergent transcription. The strand-specific measurement of Pol II density along the length of the dTRE is indicated with the red (plus strand) and blue (minus strand) boxes. The numbers of dTREs in K562 cells that occur prior to (NHS), upon acute heat shock (HS), or in both conditions are shown. **b** MA-plot showing dTREs with significantly up- or downregulated Pol II density across the length of the regulatory element. **c** Density of transcriptionally engaged Pol II at individual up- or downregulated dTREs. The dTREs are sorted by the increasing distance between the Pol II pause sites at the sense and the anti-sense strands, and the signal is centered to the middle coordinate between the pause sites

released into the gene bodies (downregulated genes) (Fig. 3b,c). Thus, rather than correlating with the gene's productive transcription per se, the histone H4 acetylation increased concurrently with the local Pol II density. In accordance, the dTREs that gained Pol II upon stress displayed increased acetylation of histone H4, whereas the dTREs that showed reduction in Pol II density maintained their status of histone H4 acetylation (Fig. 3d, e and Supplementary Fig. 3b). Intriguingly, acetylation of histone H4 spread along with the Pol II into the coding regions of the upregulated genes, indicating that the dynamic adjustment of histone acetylation accompanied the distinct steps of the transcriptional process (Fig. 3a-c and Supplementary Fig. 3c). The increase in the histone H4 acetylation at the gene body was particularly evident at the 5′ regions where Pol II, along with complexes that phosphorylate its C-terminal domain, 5′-cap the pre-mRNA, and facilitate transcriptional elongation, are at high concentrations[53].

**PIC positioning defines orientation and primes induction.** In response to stress, thousands of genes are repressed by a broadly acting mechanism that reduces the release of paused Pol

II (Fig. 1). Given such a global restraint on gene expression, we sought to determine how a subset of genes rapidly launches Pol II into elongation. Furthermore, we addressed what mechanistically defines the directionality of transcription at the divergent promoters where two distinct core initiation regions provide assembly platforms for Pol II towards the sense and anti-sense strands[19, 40, 41] (schematically presented by arrows in Fig. 1a). To address these questions, we mapped the architecture of divergent promoters prior to stress using the PRO-seq data of this study, together with GRO-cap[19] and ChIP-nexus of TATA-box-binding protein (TBP)[54], which together enable nucleotide-resolution profiling of the positioning, initiation, pausing, and elongation of Pol II. Consistent with previous reports[19, 39], transcribed genes, on average, displayed a comparable intensity of transcription machinery at the coupled initiation regions of coding and divergent strands (Supplementary Fig. 4a). However, mapping the profile of engaged Pol II selectively at genes that were either highly transcribed (RPK > 500 in NHS) or highly upregulated (log2 fold enrichment >2 and change in RPK > 200), uncovered a remarkably strong signal of Pol II complex at the promoter-proximal region of the coding strand (Fig. 4a), indicating established directionality towards the gene.

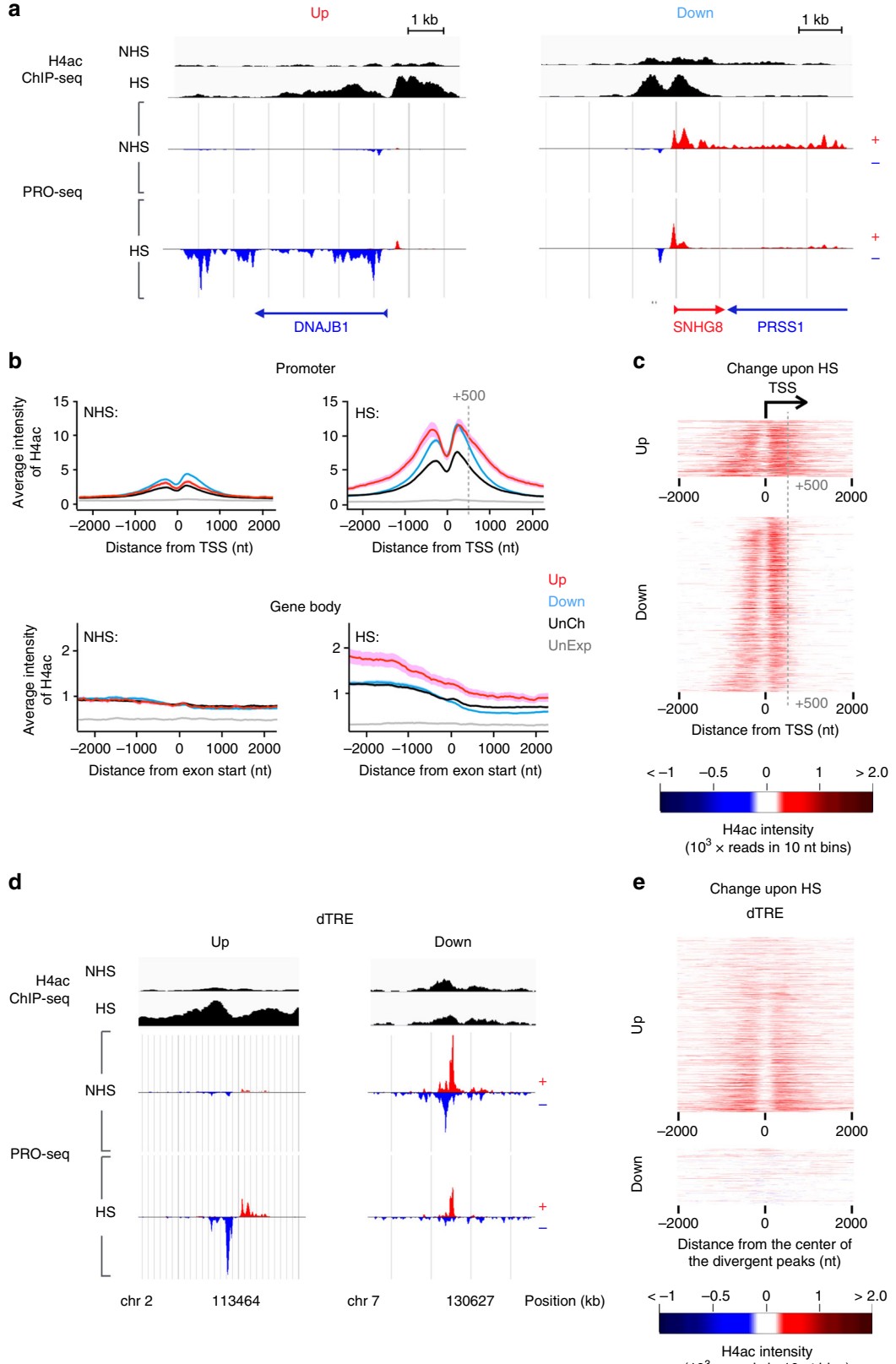

**Fig. 3** The state of histone H4 acetylation changes in cells exposed to acute stress. **a** Genome browser images of heat-induced (*DNAJB1*) and heat-repressed (*SNHG8*) genes, showing the histone H4 acetylation (*black*) and transcription from plus (*red*) and minus (*blue*) strands prior to (NHS) and upon (HS) heat stress. **b** Average ChIP-seq intensity of histone H4 acetylation at promoters (*upper panels*) and gene bodies (*lower panels*) of genes grouped by their transcriptional response. **c** The change in histone H4 acetylation upon stress at individual promoters of up- and downregulated genes. The grey dashed line in (**b**, **c**) marks the +500 nt position from TSS. **d** Genome browser images of histone H4 acetylation at dTREs with upregulated (*left*) and downregulated (*right*) Pol II density upon stress. **e** The change in histone H4 acetylation upon stress at individual up- or downregulated dTREs

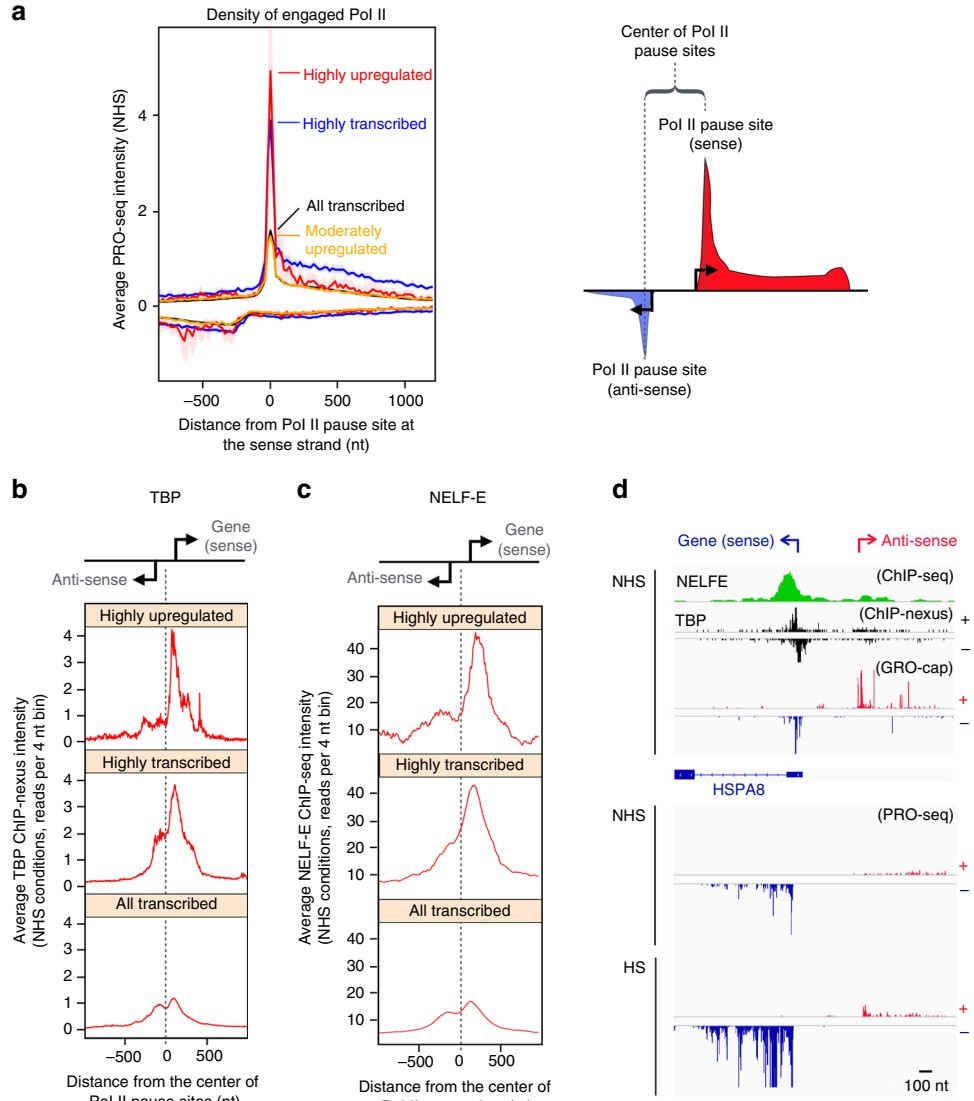

**Fig. 4** Directionality and rapid transcriptional induction is pre-wired in the promoter architecture. **a** PRO-seq profile of highly upregulated, highly transcribed, moderately upregulated and all transcribed genes, centered on the Pol II pause site at the coding strand. The intensity of transcriptionally engaged Pol II at the coding strand is indicated above the value zero, respective intensity at the non-coding strand is depicted with negative values. The right panel schematically depicts the Pol II profile at divergent regulatory elements, indicating TSSs to sense and anti-sense directions with arrows, Pol II pause sites at sense and anti-sense strands with dotted lines, and the mid coordinate between the Pol II pause sites with a bracket. **b**, **c** Average intensity of (**b**) TBP ChIP-nexus and (**c**) NELF-E ChIP-seq at highly upregulated, highly transcribed and all transcribed genes. The corresponding heatmaps are shown in Supplementary Fig. 4c. **d** Browser shot of *HSPA8* gene indicating the positioning of NELF-E and TBP with respect to the GRO-cap mapping of transcription start sites at the sense and the anti-sense strands, and the PRO-seq profile of transcriptionally engaged Pol II prior to (NHS) and upon (HS) heat stress. ChIP-nexus data set for TBP was obtained from He et al.[54], ChIP-seq for NELF-E is from the ENCODE[55], and GRO-cap from Core et al.[19]

The highly upregulated genes contained prominent levels of General Transcription Factors (GTFs)[55] (Supplementary Fig. 4b), and showed strand-specific pausing of Pol II (Fig. 4a), suggesting that the heat shock responsive promoters had assembled promoter core machinery prior to HS that efficiently recruited Pol II to the pause sites of the coding strand.

To investigate whether the directionality of transcription was established at the assembly or pausing of the transcription machinery, we mapped the positioning of TBP, which is a major component of the Pre-Initiation Complex (PIC). Analogous to the polarized Pol II positioning (Fig. 4a), the highly upregulated genes contained substantially more TBP at the core promoter of the coding strand than at the coupled core promoter of the divergent strand, as evidenced by the heatmaps (Supplementary Fig. 4c) and composite profiles (Fig. 4b) of TBP ChIP-nexus.

Comparison of ChIP-nexus and ChIP-seq profiles of TBP clearly demonstrated the higher resolution of ChIP-nexus over conventional ChIP-seq, allowing distinction between the coupled core promoters of the sense and anti-sense strands (Supplementary Fig. 4d). However, also the average ChIP-seq profiles of TBP, GTF2B and GTF2F1 (provided by the ENCODE consortium)[55] showed positional preference towards the core promoter of the sense strand, specifically at highly transcribed and highly upregulated genes (Supplementary Fig. 4d, e), supporting polarized positioning of PIC via binding of critical core components. Taken together, these results indicate that the PIC can be selectively assembled to a defined initiation region within a divergent regulatory element, and that the choice of the assembly site for Pol II complex is a major determinant of directionality.

Negative Elongation Factor (NELF) co-localizes with the paused Pol II to inhibit its release into productive elongation[56–58]. We found a strong occupancy of NELF-E subunit immediately downstream of the TSS of highly upregulated genes (Fig. 4c, d and Supplementary Fig. 4c), supporting NELF's proposed role in preventing the premature escape of Pol II into productive elongation. It is notable that abundant NELF-E levels were found also at the pause sites of highly transcribed genes (Supplementary Fig. 4c), indicating a more versatile role for NELF than a sheer block of the pause-release at inducible genes. The presence of NELF at the pause site of highly transcribed genes likely reflects the general requirement of promoter-proximal pausing to prepare the Pol II complex for elongation, and suggests that the NELF complex is important for regulation of the pause-release at poised as well as actively transcribed genes. Together, the selective positioning of the PIC, the strand-specific loading and pausing of Pol II, and the strong association of NELF with the paused Pol II complex, demonstrate how promoter architecture can establish orientation and prime rapid transcriptional responses.

**HSF1 drives the induction of primed genes**. HSF1 is known as the major *trans*-activator in heat-stressed eukaryote cells[21, 59], and it was recently shown to function primarily at the step of the pause-release[16, 18]. To determine the extent to which HSF1 orchestrated the heat-triggered transcriptional reprogramming of genes and dTREs in human cells, we compared HSF1-binding sites upon acute stress[28] with the heat-induced changes in the nascent RNA synthesis. HSF1-binding was detected at 29% of upregulated and 2% of downregulated genes, as measured from −2.5 kb from the TSS to the polyA-site (Fig. 5a), and regardless of the stress-responsiveness of the gene, HSF1-binding occurred close to the TSS (Supplementary Fig. 5a). HSF1-binding was detected at the vast majority of the highly upregulated genes (Fig. 5a) in close proximity to the paused Pol II (Fig. 5b, Supplementary Fig 5b). In accordance, the HSF1-binding intensity positively correlated (rho 0.38, *P*-value $4 \times 10^{-9}$) with the gene's heat-induction, whereas we detected no correlation (rho −0.06, *P*-value 0.48) between the HSF1-binding intensity and the magnitude of downregulation (Supplementary Fig. 5c). The highly activated genes encode a defined subset of heat shock proteins (HSPs), co-chaperones, chaperonins and polyubiquitin (Supplementary Fig. 5d, e) that have been shown in many human cells to be induced by heat stress in an HSF1-dependent manner[27, 28, 60, 61]. Downregulated genes, instead, included components of the translation machinery, which were highly transcribed in unstressed cells, but strongly repressed upon heat shock and practically devoid of HSF1-binding (Supplementary Fig. 5d, e). These results illustrate the prompt shift in the stressed cells from protein production to maintenance of the proteome quality, corroborating the previous findings by us and others that stressed cells utilize HSF1 to rapidly *trans*-activate the stress-specific repertoire of chaperone complexes. Moreover, together with the above analysis of the promoter before HS, these results uncover the tight link between the pre-assembled promoter architecture and the binding of an inducible *trans*-activator close to the paused Pol II to launch transcriptional responses to heat stimuli.

**Local chromatin architecture can restrict gene activation**. The strong *trans*-activating capacity of HSF1 at highly upregulated genes provoked us to address the mechanisms that prevented the gene activation at a subset of HSF1-bound promoters. Previously, HSF2 was reported to inhibit or modulate HSF1-driven *trans*-activation[62–64], but we could not detect HSF2

at the HSF1-bound downregulated genes. On the contrary, HSF2 accompanied HSF1 at the genes that displayed the highest heat-induction (Fig. 5c), indicating that the HSF1-HSF2 interplay occurred mainly at upregulated genes. To extend the chromatin analyses beyond HSFs, we utilized the ENCODE databases[55] and mapped the local enrichment of chromatin associated proteins, histone modifications and chromatin modifiers at HSF1-targeted loci. The applicability of the ENCODE data sets to analyses of cells cultured in our laboratories was ensured by showing the high correlation of the genome-wide transcriptional profiles of K562 cells in our study (PRO-seq at NHS) with K562 transcriptional profiles of three distinct ENCODE laboratories using previously published Pol II ChIP-seq data sets (Supplementary Fig. 6a). Considering the lower resolution, intrinsic background signal, and the lack of strand-specificity in ChIP-seq, we selected the most actively-transcribed 10,000 genes in our PRO-seq data, measured PRO-seq intensities from both strands, and collected read counts from gene bodies (+2000 nt from TSS to −2000 nt from polyA site). As indicated in Supplementary Fig. 6a, the transcriptional profiles correlated extremely well (rho 0.9) between cells cultured in distinct laboratories, and validated the usage of ENCODE data sets in this study.

Detailed mapping of chromatin associated proteins from the ENCODE databases[55] identified a distinct chromatin composition at the HSF1-bound regions that either supported or restrained heat-induction. Among the 150 functional data sets queried, particularly striking was the strong and selective presence of Specificity Factor 2 (SP2) at the HSF1-targeted promoters of downregulated genes (Fig. 5c). Indeed, SP2 displayed nearly the maximum ENCODE binding score (on the scale 0–1000; https://genome.ucsc.edu/FAQ/FAQformat.html#format12) at the HSF1-bound downregulated genes (Fig. 5d). Moreover, the summits of HSF1 and SP2 peaks were in very close proximity (Fig. 5d), and the high binding score of SP2 over HSF1 predicted low gene transcription upon stress (Supplementary Fig. 6b, c). The tight localization of SP2 between HSF1 and the complex containing paused Pol II and NELF-E (Fig. 5e) opens intriguing possibilities to be addressed in future studies on whether SP2 prevents HSF1 from gaining its *trans*-activator competence, or functions as a local insulator that restricts HSF1 from contacting the paused transcription machinery.

**HSF1 binds to heat-induced enhancers**. Most of the HSF1 target sites in yeast[22], fly[24, 25], murine[16] and human cells[27, 28] reside in intergenic regions. To explore the functional significance of HSF1-binding beyond the gene promoters, we intersected the HSF1 target coordinates with those of dTREs. Heat-induced binding of HSF1 occurred at 446 dTREs (Supplementary Data 3), indicating that active enhancers are a major class of HSF1 target sites. The HSF1-targeted dTREs were predominantly upregulated (Fig. 6a), displayed high transcriptional induction (Fig. 6b), and gained an increase in histone H4 acetylation upon stress (Fig. 6c, d). Intriguingly, the HSF1-bound dTREs that did not show significant changes in transcription also gained acetylation at histone H4 upon acute stress (Fig. 6c, d). This change in chromatin state but not transcription likely reflects functional differences of distinct enhancer classes, a subset of which could primarily work as assembly platforms for transcriptional regulators, while other classes also produce considerable amounts of eRNAs.

**Lineage-specific factors prime enhancer activation**. The rapidly induced transcription from dTREs prompted us to investigate the chromatin architecture at heat-activated dTREs. Prior to stress,

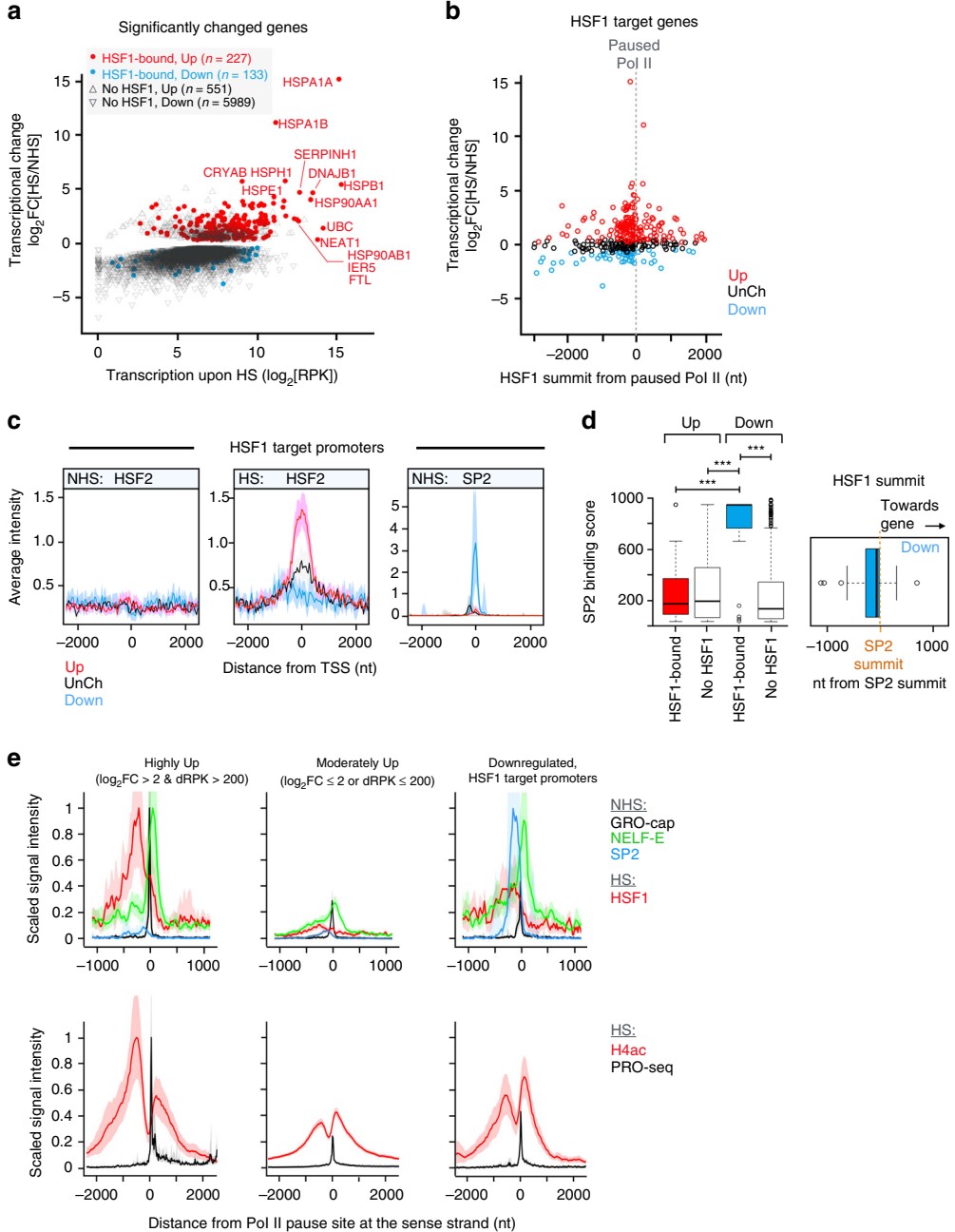

**Fig. 5** Local chromatin architecture is permissive or restrictive for HSF1-mediated *trans*-activation. **a** Transcriptional change of up and downregulated genes, plotted against the transcriptional level upon stress. The HSF1-bound upregulated genes are indicated with red, the HSF1-bound downregulated genes with blue closed circles. **b** Localization of HSF1 summit point from the Pol II pause site at the coding strand, plotted against the gene's transcriptional change upon heat shock. **c** The average occupancy of HSF2 and SP2 at HSF1 target promoters. **d** *Left panel*: ENCODE binding score (proportional number between 0–1000; https://genome.ucsc.edu/FAQ/FAQformat.html#format12) of SP2 at the HSF1-bound and HSF1-unbound upregulated (Up) and downregulated (Down) genes. Significant *P*-values (Mann–Whitney *U*-test) are shown; the three asterisks indicating values lower than 0.0005. The position of HSF1 peak summit from the peak summit of SP2 is depicted with respect to the directionality of the divergent promoter (*right panel*). **e** Scaled ChIP-seq, PRO-seq and GRO-cap intensities at indicated gene groups. The highest average signal intensity for each factor in any bin across the gene groups is used as normalization constant, bringing the maximum signal to value 1. ChIP-seq data sets for NELF-E and SP2 were obtained from the ENCODE[55], GRO-cap is from Core et al.[19]

we did not find considerable enrichment of Pol II or GTFs at the upregulated dTREs (Fig. 6b, e). Instead, lineage-specific transcriptional regulators[65–67], such as GATA-binding proteins (GATA1 and GATA2) and T-cell acute lymphoid leukemia 1 (TAL1), occupied the HSF1-bound and unbound dTREs that gained transcriptional activity upon stress (Fig. 6e and Supplementary Fig. 6d). These results are in line with a recent

study where GATA1/2 and TAL1 were shown to define the repertoire of active enhancers and the transcriptional profile during hematopoiesis[67]. Furthermore, the finding of high levels of GATA1/2 and TAL1 at heat-activated dTREs suggests an unexpected role for lineage-specific factors in priming transcriptional responsiveness of enhancers to external stimuli, as shown here in K562 cells exposed to acute stress.

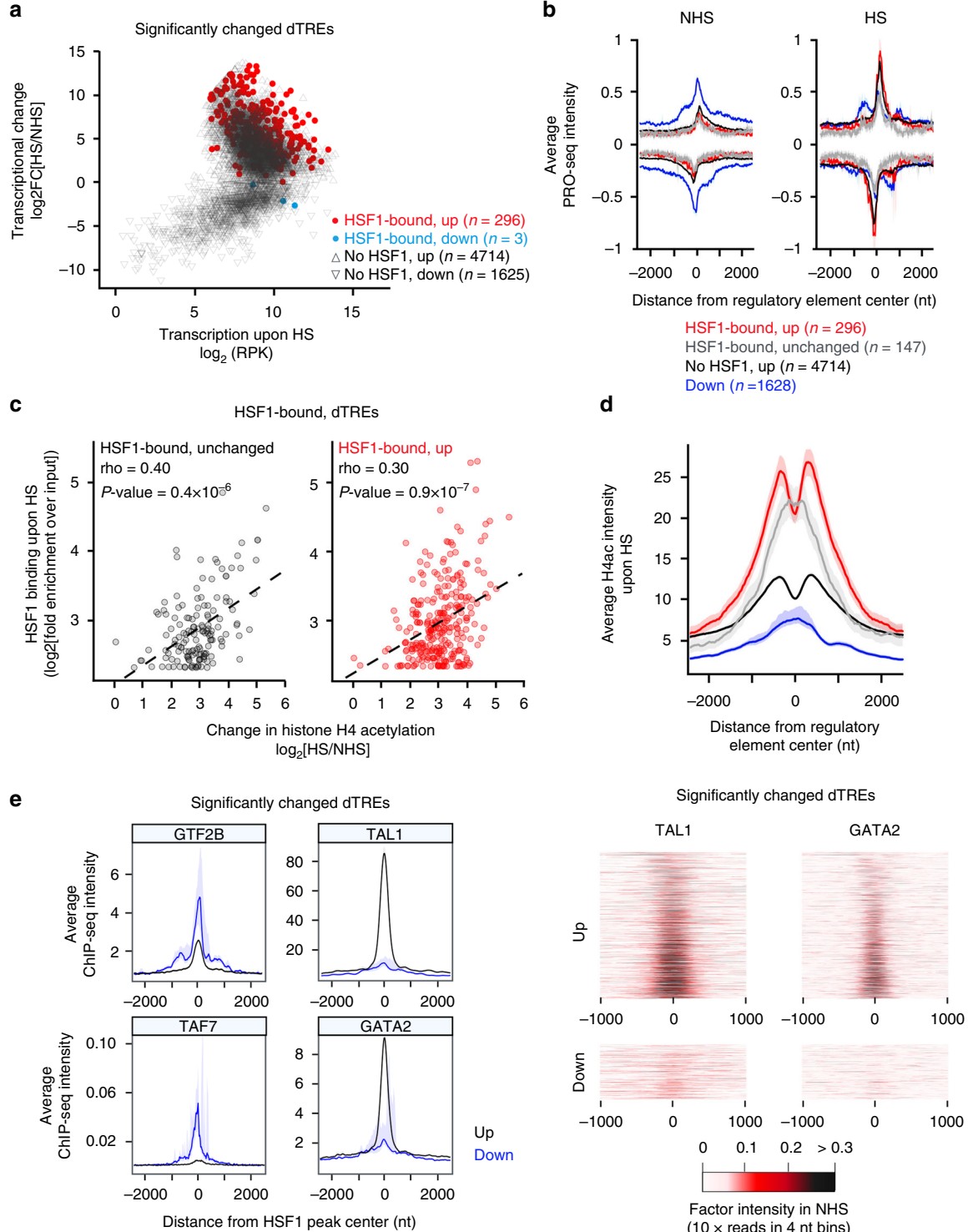

**Fig. 6** Lineage-specific transcription factors prime enhancers for transcriptional induction. **a** Transcriptional change of significantly changed dTREs plotted as function of transcription level upon stress. The HSF1-bound upregulated dTREs are indicated with red, the HSF1-bound downregulated dTREs with blue closed circles. The numbers of HSF1-targeted or un-targeted dTREs are indicated. **b** PRO-seq profile of indicated dTREs in NHS and HS conditions. **c** HSF1 binding intensity as the function of heat-induced change in the histone H4 acetylation at individual dTREs. The coefficients (rho) and *P*-values are according to Spearman's rank correlation. The correlation lines are fitted for linear regression. **d** Average histone H4 acetylation at indicated dTREs upon HS. **e** Left panels: Average ChIP-seq intensity of GTF2B, TAF7, TAL1 and GATA2 at up- and downregulated dTREs. Right panels show heatmaps of TAL1 and GATA2 at up- and downregulated dTREs. ChIP-seq data sets for GTF2B, TAF7, GATA2 and TAL1 are from the ENCODE[55]

**HSF1 binding and histone acetylation at untranscribed loci**. A considerable fraction (40%) of HSF1 target sites localized neither to promoters nor to dTREs (Fig. 7a). These HSF1 target sites contained HSEs but did not initiate transcription under either

optimal growth conditions or upon a 30-min heat shock (Fig. 7a), and hence we termed them untranscribed HSF1 target regions. Deduced from the ENCODE databases[55], the HSF1-targeted promoters and dTREs resided in open chromatin conformation

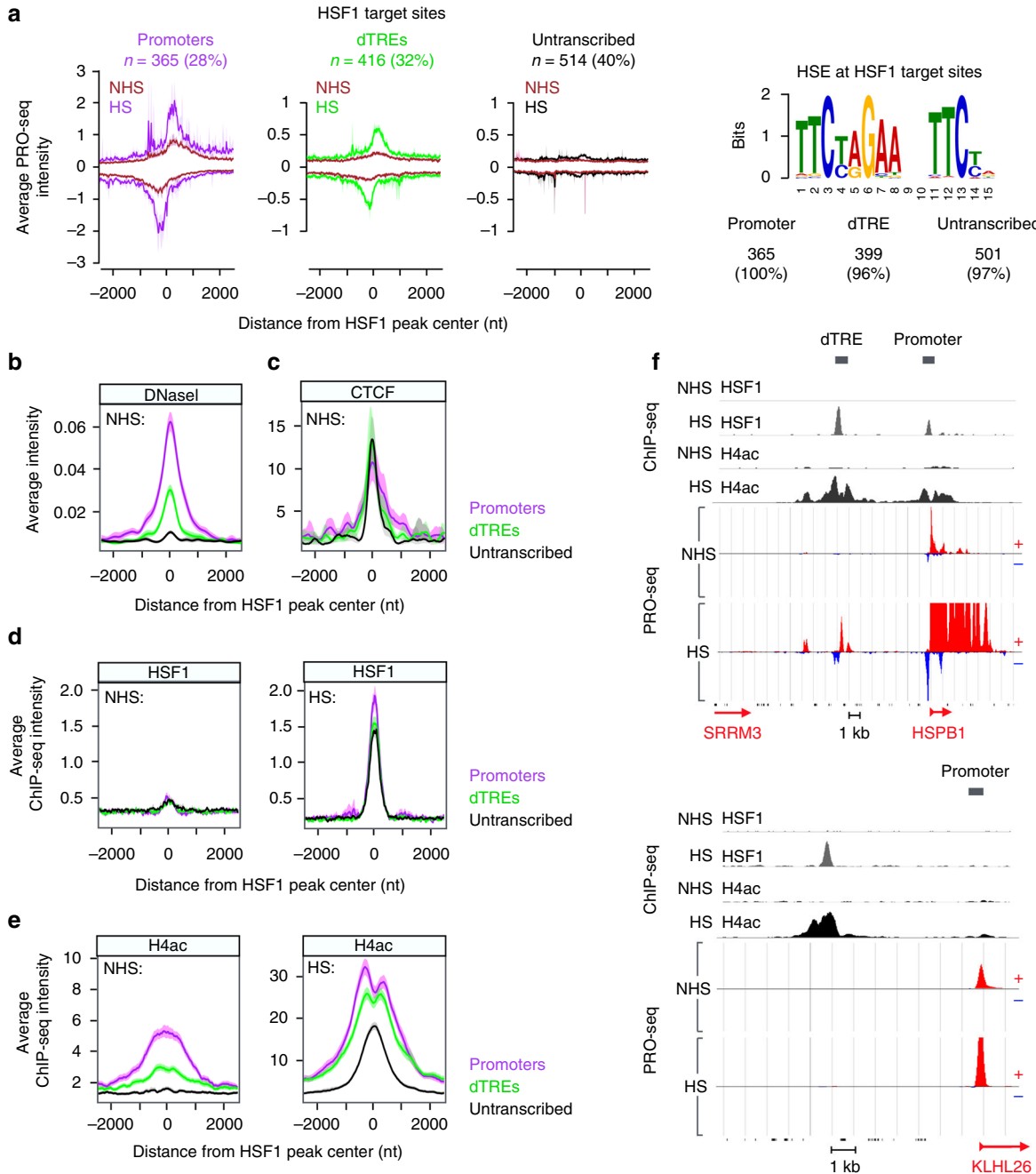

**Fig. 7** HSF1 binds to regions that are devoid of initiating Pol II but gain hyper-acetylation at histone H4 upon stress. **a** *Left panels*: Average Pol II density at the HSF1-bound promoters, dTREs and untranscribed genomic regions. N indicates the number, and percentage the share, of HSF1 target sites in respective category. The *right panel* shows the number and percentage of HSF1 target sites that contain a MEME-ChIP identified HSE (Supplementary Methods). **b–e** Average intensities of (**b**) DNaseI sensitivity, (**c**) CTCF ChIP-seq, (**d**) HSF1 ChIP-seq and (**e**) histone H4 acetylation ChIP-seq at HSF1-targeted promoters (*purple*), dTREs (*green*) and untranscribed regions (*black*). **f** Genome browser examples of a promoter and a dTRE (*upper panel*), and an untranscribed genomic region (*lower panel*) that are bound by HSF1 upon heat shock. HSF1 binding (*gray*), histone H4 acetylation (*black*) and transcriptionally engaged Pol II (*red* and *blue*) are shown in NHS and HS conditions. Data sets of DNaseI sensitivity and CTCF ChIP-seq (Broad institute) were obtained from the ENCODE[55]

and contained clusters of transcriptional regulators and chromatin remodelers already prior to stress (Fig. 7b and Supplementary Fig. 7a, b). In stark contrast, the untranscribed HSF1 target sites localized to closed chromatin regions, which were devoid of transcriptional regulators, chromatin modifiers, and histone modifications that are typically associated with open chromatin (Fig. 7b and Supplementary Fig. 7a, b). The HSF1-targeted untranscribed regions did display a considerable

amount of the CCCTC-binding Factor (CTCF) and, to a lesser extent, components of the cohesin complex (Fig. 7c and Supplementary Figs 7 and 8). These chromatin-associated architectural proteins have been shown to organize genomes into functional compartments and provide topology for transcription factor binding[68–72].

The closed chromatin did not prevent HSF1 from rapidly gaining access to the HSEs (Fig. 7d). Indeed, concurrent

hyper-acetylation of histone H4 hallmarked HSF1-binding both at open and closed chromatin, demonstrating the profound changes in the chromatin state at HSF1-targeted regions (Fig. 7e, f). Besides suggesting a rapid increase in chromatin accessibility upon stress, the robust histone H4 acetylation at untranscribed regions demonstrated that histone-acetylation can be kinetically separated from, or occur regardless of, the transcriptional engagement of Pol II (Fig. 7f). Likewise, the finding that HSF1 localized to regions that did not contain engaged Pol II, revealed that the binding of a potent *trans*-activator can occur without pre-existing or concurrent loading of the transcription machinery. In conclusion, HSF1 targeted both transcription-primed and CTCF-primed chromatin that gained localized, heat-induced acetylation, indicative of assembly of transcription factors to pre-existing transcription-primed or architectural protein-primed platforms at genes and intergenic regions.

## Discussion

The human genome encodes a great number of transcripts whose expression needs to be precisely coordinated in space and time. In this study, we took the advantage of the immediate transcriptional change that is provoked by heat shock and monitored at high sensitivity the synthesis of long and short, stable and unstable RNAs that arise from genes and distal regulatory elements. Deciphering the rapidly induced transcriptional response in the context of the chromatin architecture revealed that: (1) Cells elicit a massive response to stress by reprogramming transcription and adjusting the chromatin landscape of genes and enhancers. (2) A gene's response to stress is defined at the pause-release of promoter-proximal Pol II. (3) Strand-specific assembly of PIC and paused Pol II sets up the orientation of promoters and pre-wires transcriptional activation. (4) Lineage-specific factors prime enhancer transcription. (5) Potent *trans*-activator HSF1 localizes to transcription-primed and architectural protein-primed genomic regions. (6) Local SP2-associated chromatin architecture can restrain the activation of primed, HSF1-targeted gene promoters.

We propose a model (Fig. 8), where coordination of the single step of promoter-proximal pause-release determines the whole transcriptional response of the human genome in high fidelity. In particular, inhibition of the pause-release causes transcribing Pol II to clear off from thousands of downregulated genes, providing free Pol II for rapid loading into available initiation regions. At upregulated genes, the pre-assembled PIC at the core promoter of the coding strand directs Pol II towards the gene, coupling release of the Pol II to instant filling of the freed pause site. The global transcriptional reprogramming is likely to involve Positive Transcription Elongation Factor b (P-TEFb), which phosphorylates NELF and CTD of Pol II, releasing the paused Pol II into productive elongation[53]. The inhibition of P-TEFb could be counteracted by strong *trans*-activators, such as HSF1, the activation of which has been coupled with recruitment of P-TEFb to heat-activated genes[73].

HSF1 is one of the most potent *trans*-activators known, and it is capable of inducing transcription to a level where Pol II is tightly packed on genes. We found strong HSF1-binding at the heat-activated genes, but also uncovered new classes of HSF1 target sites occurring at dTREs and untranscribed regions within closed chromatin. Characteristic for HSF1-binding was the simultaneous acetylation of histone H4, indicative of the complex changes that occurred on activated genomic regions in the presence or absence of Pol II. Recently, HSF1 was demonstrated to promote the release of Pol II into elongation[16, 18], which

provides a mechanistic explanation for why the poised gene promoter and the well-positioned binding of HSF1 upstream of the paused Pol II rapidly launched transcription. At certain gene promoters HSF1-binding did not enforce transcriptional activation. Among the multitude of functional genomic data sets queried, SP2 emerged as the most distinctive factor that occupied HSF1-bound downregulated genes and correlated with the low transcription in stressed cells. Intriguingly, the HSF1-bound promoters of downregulated genes were devoid of HSF2. The recently solved crystal structure of the HSF2 DNA-binding domain revealed that the amino acids that are in direct contact with the HSE are highly conserved among HSFs, while the amino acids that face away from the DNA backbone are diverse[74], which mechanistically could explain why HSF1 and HSF2 bind to practically identical HSEs but mediate profoundly different actions at the level of chromatin[28]. To this end, the selective presence of SP2, and the lack of HSF2, at the downregulated HSF1 target genes emphasizes the importance of the local chromatin environment and interactions of transcription factors beyond the DNA sequence for determining the factor-specific recruitment and the transcriptional outcome thereof.

Examining the simultaneous change in the chromatin state and transcription uncovered that hyper-acetylation of histone H4 accompanied the increase in Pol II density at promoters and dTREs. Intriguingly, islands of histone H4 acetylation appeared also at sites that did not initiate transcription, suggesting that distal Untranscribed Regulatory Elements (dUREs) are formed upon heat shock (Fig. 8). The emergence of a new repertoire of enhancers and untranscribed islands of histone H4 acetylation manifests the massive reshaping of the regulatory element landscape in stressed cells, and highlights the dynamic interplay of genes, distal regulatory elements, and the chromatin architecture. In conclusion, our analyses of the dynamic transcriptional tuning of genes and dTREs reveal how the local chromatin architecture primes transcriptional responses, and how genes, enhancers and the chromatin state are coordinated across the human genome.

## Methods

**Cell lines and heat shock treatment**. Human K562 erythroleukemia cells were maintained at 37 °C in a humidified 5% CO$_2$ atmosphere and cultured in RPMI medium (Sigma), containing 10% fetal calf serum, 2 mM L-glutamate, 100 µg ml$^{-1}$ streptomycin, and 100 U ml$^{-1}$ penicillin. The K562 cells originated from ATCC, were tested to be mycoplasma free, and displayed morphology, proliferation rate and transcriptional profile characteristic to K562 cells. To avoid provoking transcriptional changes by freshly added media, the cells were expanded 24 h prior to the treatments to confluence of 2 × 10$^5$ cells per ml. For treatments, 2 × 10$^7$ cells per sample were concentrated in 10 ml culture media and placed in 37 °C incubator (NHS) or a 42 °C water bath (HS) for 30 min. In the water bath, the temperature of the cell suspension reached 42 °C within 4 min.

**ChIP-seq**. Immunoprecipitation of chromatin with acetylated histone H4 was conducted as previously described[28], using an antibody that recognized a broad range of histone H4 acetylations (Upstate 06–866). In detail, the cells were cross-linked in 1% formaldehyde for 5 min on ice, quenched in 125 mM glycine for 5 min, washed twice with ice cold PBS, and flash frozen. The cells were lyzed in ChIP lysis buffer (1% SDS, 10 mM EDTA, 50 mM Tris pH 8.0, 1× protease inhibitor cocktail from Roche), and the chromatin was sheared to 200–500 bp fragments by 10-min sonication (Bioruptor, Diagenode), using high throughput settings with 30 s on/30 s off cycles. The lysate was pre-cleared with protein G-coated sepharose beads (Amersham Biosciences), 4 µl of Histone H4ac (Upstate 06–866) antibody incubated over night at 4 °C, and the immunocomplexes washed twice in wash buffers 1 (0.1% SDS, 1% Triton X-100, 2 mM EDTA, pH 8.0, 150 mM NaCl, 20 mM Tris-HCl, pH 8.0), 2 (0.1% SDS, 1% Triton X-100, 2 mM EDTA, pH 8.0, 500 mM NaCl, 20 mM Tris-HCl, pH 8.0) and 3 (20 mM Tris-HCl, pH 8.0, 1 mM EDTA, pH 8.0, 10% glycerol). Proteins were degraded with Proteinase K, and RNA with RNaseA, and cross-links were reversed over nigh at 65 °C. DNA was purified twice with phenol:chloroform and once with chloroform, and precipitated using EtOH. Ten biological ChIP replicates of non-treated and heat-treated samples, respectively, were pooled and purified with Qiaquick DNA purification columns, and the libraries were generated with Nextera DNA Sample Preparation

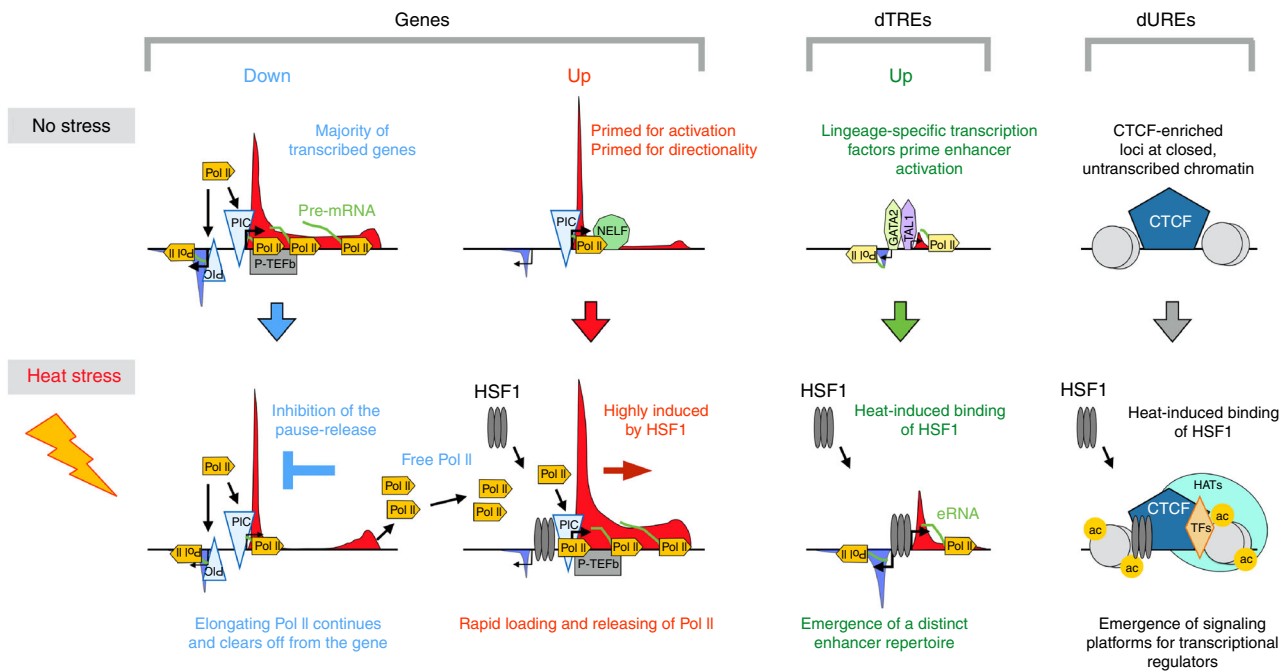

**Fig. 8** Rapid and coordinated reprogramming of genes and distal regulatory elements in stressed human cells. Model showing the rapid heat-induced response of the human genome, including reprogramming of gene transcription and establishment of a stress-specific repertoire of distal regulatory elements. Inhibition of the pause-release of promoter–proximal Pol II clears transcription complexes from the downregulated genes, elevating the concentration of free Pol II in the cell. The heat-induced genes are primed for directionality and rapid activation by the pre-assembled PIC on the core promoter of the coding strand, and by the paused Pol II at the 5′ region of the gene. Upon stress, *trans*-activators, such as HSF1, launch Pol II from the primed genes into productive elongation. The elevated levels of free Pol II and the highly positioned PIC on the coding strand enable instant loading of Pol II to the freed pause sites, efficiently launching rounds of transcript synthesis from the activated genes. The free Pol II allows also tuning up of the enhancer repertoire, increasing eRNA-production at dTREs that are marked by lineage-specific transcription factors, for example GATA1, GATA2 and TAL1 as shown in this study. In addition to reprogramming TREs, heat stress causes emergence of putative untranscribed regulatory elements, as demonstrated by the acetylation of histone H4 and recruitment of HSF1 to CTCF-rich loci that do not contain components of Pol II complex. Only the key regulatory factors discussed in the text are shown. ac, acetylation; CTCF, CCCTC-binding factor; Down, downregulated genes upon acute heat stress, dTRE, distal transcription regulatory element; dURE, distal untranscribed regulatory element; eRNA, enhancer RNA; GATA, GATA-binding protein; HSF1, heat shock factor 1; PIC, pre-initiation complex; P-TEFb, positive transcription elongation factor b; TAL1, T-cell acute lymphoid leukemia 1; TF, transcription factor, Up, upregulated genes or dTREs upon acute heat stress

kit (Illumina). Total fragmented chromatin (Input) was used as a normalization control. Sequencing was performed with HiSeq2500 (Illumina).

The sequenced ChIP-reads were trimmed, clipped and filtered with fastx toolkit (http://hannonlab.cshl.edu/fastx_toolkit/) to 36 nucleotides with a minimum requirement of 99% probability for correct identification for at least 0.8 fraction of bases in a read. The reads were aligned to the human genome 19 (GRCh37/hg19) with Bowtie[75], and the peaks were called with the MACS 1.4 software[76] using total fragmented chromatin as control. The complete raw data for histone H4 acetylation is available at Gene Expression Omnibus (GEO) database (http://www.ncbi.nlm.nih.gov/geo) under accession: GSE89382. See Supplementary Methods for information on visualization of genome-wide ChIP-seq data sets.

**PRO-seq**. PRO-seq was conducted as previously described[11, 77]. In detail, the untreated or heat-treated samples were washed with PBS, and incubated for 5 min in nuclear isolation buffer (10 mM Tris-Cl pH 8.0, 300 mM sucrose, 3 mM CaCl₂, 2 mM MgAc₂, 0.1% TritonX-100, 0.5 mM DTT), followed by 25x douncing (Wheaton, #357546, loose pestle), to isolate the nuclei. After centrifugation (1000 g, 5 min, 4 °C), the nuclei were collected and flash frozen in storage buffer (10 mM Tris-HCl pH 8.0, 25% glycerol, 5 mM MgAc₂, 0.1 mM EDTA, 5 mM DTT). The nuclear run-on reactions were performed at 37 °C for 3 min in the presence of 0.05 mM biotin-A/C/G/UTP (Perkin Elmer), 0.5% sarkosyl, 5 mM Tris-HCl (pH 8.0), 1.5 mM MgCl₂, 0.5 mM DTT, 150 mM KCl and 0.1 units per ul Superase RNase inhibitor (Life Technologies). Total RNA was isolated using Trizol LS (Life Technlogies), pelleted by EtOH-precipitation and base hydrolyzed with NaOH to 100–150 nt fragments. For high specificity, the biotinylated nascent transcripts were purified three times with streptavidin-coated magnetic beads (Life Technologies), each round followed by isolation with Trizol (Life Technlogies) and EtOH-precipitation. The 5′ cap of transcripts was removed with tobacco acid pyrophosphate (Epicentre) and the 5′ hydroxyl group repaired with T4 polynucleotide kinase (BioLabs). The libraries were generated using TruSeq small-RNA adaptors and sequenced using HiSeq2500 (Illumina).

The PRO-seq reads were adapter-clipped using cutadapt[78] and trimmed and filtered to 15–36 bp with fastx (http://hannonlab.cshl.edu/fastx_toolkit/). Reads that did not map to ribosomal RNA genes were aligned to hg19 using Bowtie[75] selecting only uniquely mapping reads with up to two mismatches. The complete raw data for PRO-seq in human K562 cells is available at GEO database (http://www.ncbi.nlm.nih.gov/geo) under accession: GSE89230.

**Data analysis of aligned Pro-seq reads**. To analyze transcription of genes, we filtered RefGen database to obtain a non-redundant list of RNA- and protein-coding genes with 500 nt or more in length ($n = 23,698$). The density and location of transcriptionally engaged Pol II at each gene's promoter–proximal region (−100 to +400 nt from TSS) was defined as reads per kb mappable genomic DNA (RPK) at a 50-nt window with the highest read count. The localization and intensity of Pol II at the divergent strand, was scored at the 50-nt window from −800 to +100 region from the TSS. To measure the transcription of a gene's coding region, the average Pol II density (RPK) across each gene body (+500 nt from the TSS to −500 nt from the polyA site) was measured. The correlation of the two biological replicates in each condition was analyzed using Spearman correlation and plotted as density plots of reads mapping to promoter–proximal regions, gene bodies and dTREs. The reads of the two replicates highly correlated, allowing their combination for further analyses. To compare transcription in NHS versus HS conditions, each PRO-seq data set was normalized, as described and tested previously[16], using the 3′ end of long (>150 kb) genes as reference genomic regions where no transcriptional change upon a 30-min heat stress was detected. See Supplementary Methods for information on visualization of genome-wide PRO-seq data sets.

**Genes with significant transcriptional change upon stress**. For identification of genes with significantly changed transcription upon stress we utilized DESeq2, which uses the variance in biological replicates to assess significant changes between stressed and unstressed data sets[79], setting a maximum accepted *P*-value

to 0.001 and a minimum fold change (FC) 1.25 as cutoffs for calling a significant expression change. Genes that were identified changed with low confidence, constituted mainly of lowly transcribed genes, and were not considered in the downstream analyses. To remove genes whose expression change was called significant due to the activity of adjacent or partially overlapping genes (run-over transcription or internal TSS), or due to changed transcription of internal regulatory element(s), we utilized dREG[20]. dREG is a machine-based learning method that identifies transcribed regulatory elements from the human genome using support vector regression. To call a gene significantly changed, unchanged or unexpressed, we required it to have a dREG-identified TRE (minimum score 0.7), at the TSS.

**Identification of TREs in high-resolution and sensitivity**. To identify active promoters and transcribed enhancers we used the dREG software program[20] to map the broad locations of TREs. Next, we developed a novel strategy called dREG-HD that refines the coordinates of TREs using the peaks of divergently oriented Pol II. The source code of dREG-HD is available at https://github.com/Danko-Lab/dREG.HD. Briefly, we developed an epsilon-Support Vector Regression (SVR) with a Gaussian kernel to predict the level of DNaseI hypersensitivity, which peaks between the divergently oriented paused Pol II, using PRO-seq data. The SVR was trained on randomly selected positions within peaks identified by dREG extended by 200 nt on either side. To optimize model free parameters, we maximized the Pearson correlation between the imputed and experimental DNaseI[55] score on holdout sites not used during training. After the initial parameter adjustment, the dREG-HD model was trained using DNaseI[55] and PRO-seq[19] data in the whole genome in K562 cells. Next, we identified peaks in the imputed DNaseI hypersensitivity profile by fitting the imputed DNaseI signal using a cubic spline and identifying local maxima. We optimized two free parameters that control the (1) smoothness of spline curve fitting, and (2) threshold on the imputed DNaseI signal intensity. Parameters were optimized using grid optimization to achieve an appropriate trade-off between False Discovery Rate (FDR) and sensitivity on the K562 data set. Applying the TRE-caller on a GRO-seq data set, which inherently has a lower resolution, obtained from a different cell line (GM12878)[19], completely held out during model training, and resulted in 82% sensitivity for identification of DNaseI peaks within dREG sites at a 10% FDR.

**Classifying TREs into promoters and dTREs**. TREs were classified into promoters and dTREs by predicting the stability of divergent transcripts, a strategy introduced by Core and co-workers[19]. Briefly, by comparing the signal intensities of 5' capped RNAs from GRO-cap (reports both stable and unstable transcripts) with that obtained by Cap Analyses of Gene Expression (CAGE; reports stable RNAs only), Core et al.[19] grouped TREs into promoters and dTREs. While promoters produce at least one stable transcript from the two possible orientations, dTREs are defined to produce rapidly degraded divergent transcripts in both directions. The transcription profile from a divergent promoter is schematically depicted in Fig. 1a, and a respective profile from a dTRE is illustrated in Fig. 2a.

We trained a Support Vector Machine (SVM) classifier to predict whether each TRE encoded at least one stable transcription unit (promoter) or whether both of the transcripts were unstable (dTRE) using PRO-seq data and CpG/GC content[55] as input. We used the subset of dREG-HD peaks that intersect with GRO-cap pairs[19] as training examples. We separately trained and optimized independent classifiers based on the PRO-seq profile and CpG/GC content. The PRO-seq data was initially summarized as read counts in fixed non-overlapping windows centered on the dREG-HD site. First the number of bases passed to the classifier was optimized using a fixed 50 bp non-overlapping window size, and subsequently, the resolution was tuned to the total genomic area covered. We also passed the classifier both GC and CpG content in a single window centered on the dREG-HD site. Free parameters for PRO-seq classifier (window size and number of windows), as well as for CpG/GC classifier (the window size) were defined using five-fold cross-validation to optimize the Area Under the receiver operating characteristic Curve (AUC). After optimization of the free parameters, we trained a single classifier to integrate both PRO-seq and CpG/GC content. All SVM training tasks used an internal ten-fold cross-validation to obtain optimized SVM model parameters (γ of the Gaussian kernel and cost value C) using R package e1071. In a held out test set the final classifier achieved an AUC of 0.918, which is higher than achieved using the commonly used log ratio of histone modifications (log [H3K4me3/H3K4me1] at ± 2000 nt window) that obtained an AUC of 0.875. The clear outperforming of the standard method for enhancer prediction, demonstrated the high sensitivity, specificity and resolution that we obtained by calling dTREs from the nucleotide-resolution profile of PRO-seq.

**High-resolution identification of dTREs**. We used our dREG-HD classifier for high-resolution identification of TREs in NHS and HS conditions, sorting them to active promoters and dTREs. In the very rare cases when a dTRE fell on an annotated TSS (identified from RefGen), that region was not considered as a distal regulatory element. The occurrence of dTREs in NHS and HS were compared by bedtools[80], using 50% overlap as an intersect criterion for an element occurring in both conditions. For each dTRE, the average read count (RPK) and the position of the highest Pol II density in a 50-nt window were separately scored at plus and

minus strands along the whole length of the element. In heat maps the dTREs were sorted by increasing distance between the highest Pol II densities at the plus and minus strands. To identify dTREs with significantly induced or reduced Pol II density we compared, in a strand-specific manner, the PRO-seq profiles across the whole length of the dTRE in NHS versus HS. dTREs that were called significantly changed by DESeq2 (maximum accepted P-value 0.05 and a minimum FC 1.25) at either or both of the strands were called as up- or downregulated. dTREs where neither strand displayed high confidence change in Pol II density upon stress were called unchanged.

**Characterization of HSF1 and HSF2 target sites**. To analyze the recruitment of HSF1 and HSF2 to the genome upon acute stress, we used the published data sets[28] (GEO: GSE43579) that have been generated in the same cells and under the same conditions as the PRO-seq and ChIP-seq experiments reported in this study. HSF1-binding on genes was measured along the length of the gene from −2500 nt from the TSS to the polyA site. To sort HSF1 target sites to promoters and dTREs, we intersected the HSF1 peak coordinates with those of dREG-HD-identified TREs, requiring a minimum of one nucleotide overlap. The HSF1 target sites that did not occur on either promoters or dTREs were termed untranscribed. The statistical analyses on HSF1-binding intensity and the transcriptional change, or the change in histone H4 acetylation were conducted using Spearman's rank correlation.

To ensure the applicability of ENCODE data sets for analyses of chromatin composition at HSF1 target sites, the PRO-seq profile of transcription in NHS in our K562 cells was compared to Pol II ChIP-seq data in K562 cells generated by the ENCODE laboratories[55]. Considering the lower resolution, intrinsic background, and lack of strand-specificity in ChIP-seq, we selected 10,000 most actively transcribed genes that were at least 5000 nt in length (each having gene body RPK over 10 in PRO-seq), counted PRO-seq reads from both strands, and measured the total read count of PRO-seq and ChIP-seq from gene body (+2000 nt from TSS to −2000 nt from polyA site). The statistical analyses performed using Spearman's rank correlation showed excellent agreement, ensuring that our K562 cell line was behaving like the K562 cell line used by the ENCODE.

To compare HSF1-binding intensity with that of SP2, the HSF1-binding (measured over input) was first converted to ENCODE binding score, giving a spread of scores between 1000 and 107, which is according to the ENCODE guidelines. The statistical analysis of SP2 over HSF1 (SP2 ENCODE binding score/HSF1 ENCODE binding score) versus the gene's transcription upon stress was conducted with Spearman's rank correlation. The statistical significance of the difference in SP2 binding score on HSF1-bound or HSF1-unbound up- or downregulated genes was analyzed using Mann–Whitney U-test.

**Quantification of factor intensities at genomic loci**. The composite profiles and heat map analyses were generated using the bigWig package (https://github.com/andrelmartins/bigWig/) that queries the total normalized reads at defined positions. The average intensities in composite profiles, and the region-specific intensities in heat map analyses were queried in 20-bp and 10-bp bins, respectively, unless otherwise indicated. The bootstrap estimates in the composite profiles display 12.5–87.5% interval for each group. The queried genomic sites and nucleotide ranges are indicated in each figure. For mapping against exon start sites, the first and last exons were omitted from the analyses due to increased possibility for an alternative TSS and polyA site, respectively. The highly upregulated genes were selected by requiring log2 fold change (HS/NHS) > 2 and change in the RPK (HS-NHS) > 200. The moderately upregulated genes are called significantly induced by DESeq2 but do not meet the above-mentioned criteria. Highly transcribed genes displayed transcription of >500 RPK in NHS conditions and all transcribed genes were any genes that showed transcriptional activity in human K562 cells in NHS conditions.

**Composite profiles with scaled factor intensity**. To compare the level and positioning of a given factor between different gene groups, the highest average intensity in any bin of the given groups of genes was set to value 1 and every other bin normalized against this maximum value.

**Analyses of TBP ChIP-nexus data**. The bigWig data sets of TBP ChIP-nexus[54] in K562 cells were downloaded from the GEO (GSE55306). Heat maps were generated by separately querying the plus and minus strand read counts in each 4-bp bin, after which the pair of plus and minus strands were displayed in conjunction in the heat map. For metaprofiles, the plus and minus strand values were combined.

**Code availability**. Computational analyses have been performed using R and Python languages. Custom made scripts can be made available upon request.

**Data availability**. The complete raw data sets for histone H4 acetylation (GSE89382), and nascent RNA synthesis (GSE89230) are publically available at GEO databases (http://www.ncbi.nlm.nih.gov/geo). All other data that support the findings of this study are available from the corresponding authors upon request.

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

## Acknowledgements

We thank the members of the Sistonen and the Lis laboratories for valuable advice during the manuscript preparation. This work was financially supported by the Sigrid Jusélius Foundation (A.V., L.S.), Svenska Tekniska Vetenskapsakademin i Finland (A.V.), South-West Finland's Cancer Foundation (A.V.), Joe, Tor and Pentti Borg Memory Foundation (A.V.), Åbo Akademi Research Foundation (A.V.), Academy of Finland (L.S.), Finnish Cancer Organizations (L.S.), Magnus Ehrnrooth Foundation (L.S.), Åbo Akademi University (L.S.), and NIH grant RO1-GM25232 (J.T.L.). The content is solely the responsibility of the authors and does not necessarily represent the official views of the National Institutes of Health.

## Author contributions

A.V., J.T.L. and L.S. conceived and designed the study. A.V. and D.B.M. prepared the PRO-seq, and A.V. the ChIP-seq libraries. A.V., D.B.M. and M.J.G. performed the computational data analyses. T.C. and C.G.D. developed dREG-HD and performed the TRE identification and classification. A.V., D.B.M., T.C., C.G.D., J.T.L. and L.S. interpreted the results. A.V., J.T.L. and L.S. wrote the manuscript with edits from D.B.M., M.J.G., T.C. and C.G.D.

## Additional information

**Competing interests:** The authors declare no competing financial interests.

