## [Peer Review File · Nature Communications]

Reviewers' comments:

Reviewer #1 (Remarks to the Author):

The authors present a very interesting body of work addressing a central question in the stress signaling field—how might a cell coordinate the simultaneous up regulation of heat shock loci while down regulating the transcriptional activity of the majority of genes. Here, Sistonen and coworkers provide evidence of an elegant yet simple mechanism to coordinate both events that is the inhibition of the pause release of Pol II from nonstress promoters thereby creating a pool of free Pol II that is utilized at heat shock responsive promoters. Supporting the activities of heat shock genes are pre-assembled PICs (inferred based upon a previous study mapping the genomic positions of TBP (He et al. 2015) along with activated transcription factors such as HSF1. Intriguingly, the authors find that binding of HSF1, which does not recruit Pol II, correlated with SP2 occupancy suggesting SP2 might act as an insulator or directly block the ability of HSF1 nucleate Pol II. Overall, it is a well-done and exciting body of work.

1. Throughout the manuscript the authors make reference to the idea that they observe changes in the chromatin architecture that are helping to guide the heat shock transcriptional response. The abstract states “features of chromatin architecture that set the priming and control the activation..” yet little direct evidence is provide to substantiate this claim. While the authors do map where acetylated H4 is along the genome, this is just a posttranslational modification and might not represent an actual change to the chromatin architecture. Either the authors should provide a direct readout of chromatin architecture (minimally nucleosome presence/spacing could be addressed) or the text should be modified to simply refer to the H4 acetylation pattern. Statements such as “Besides uncovering the rapid opening of closed chromatin state upon stress...” should not be made unless direct evidence is provided.

2. The authors often make use of the prior published genomic maps of the genomic binding patterns of different DNA binding proteins, which is a good and appropriate use of the public data. However, for the proteins they find critical to their model the authors should validate that these factors are behaving in their cell line as expected. For example, by ChIP-PCR do the authors find the expected patterns of occupancy of TBP, CTCF, and SP2 at a few different sites?

3. The authors report on a potential role of SP2 as silencer of HSF1 transcriptional activity. Further characterization of the interaction between SP2 and HSF1 would considerably strengthen this point. For example, does inclusion of an SP2 site in an HSE-reporter lead to silence regardless of positioning or is it position dependent (e.g., does the SP2 site need to be between the HSE and start site)? Do SP2 and HSF1 physically interact?

Reviewer #2 (Remarks to the Author):

The authors have carried out an extensive study of the heat shock response in human cells and summarize the results and conclusions in a beautifully written manuscript. This is not just another global heat shock response paper. The analysis of genes, transcribed enhancers, and non-transcribed enhancers by PRO-Seq, ChIP-Seq is for the most part very clear and interesting. This is the authors description of their findings:

- 1) Cells elicit a massive response to stress by reprogramming transcription and adjusting the chromatin landscape of genes and enhancers.
- 2) A gene's response to stress is defined at the pause release of promoter-proximal Pol II.
- 3) Strand-specific assembly of PIC and Pol II sets up the orientation of promoters and pre-wires transcriptional activation.
- 4) Lineage-specific factors prime enhancer transcription.
- 5) Potent trans-activator HSF1 localizes to transcription-primed and architectural protein-primed genomic regions.
- 6) Local SP2-associated chromatin architecture can restrain the activation of primed, HSF1-

targeted gene promoters.

All of these points are important and well supported by the data.

Specific comments:

1. In the abstract the term "of the coding strand" is used. I think the authors mean "for the sense direction"

2. I didn't understand the graph of TBP in Fig. 4b. Why is TBP downstream of the paused Pol II in the sense direction. Sign seems backwards.

Overall, I really enjoyed reading this manuscript. I want my friends to read it too!

Point-by-point response:

We sincerely thank both Reviewers for their positive attitude towards our study and their valuable comments that have helped us to strengthen the manuscript. Below, please find the point-by-point response to all the concerns raised. The Reviewers' comments are indicated with *Italics*, and are followed by our response to each comment. Please note, that the revised manuscript has also been formatted to meet the lay-out requirements of Nature Communications, which in the case of this manuscript consists of shortening of the subtitles in the Results and Materials and Methods sections, reformatting the references, and condensation of the Discussion. All the changes made are indicated in the revised manuscript.

Reviewer #1

The authors present a very interesting body of work addressing a central question in the stress signaling field—how might a cell coordinate the simultaneous up regulation of heat shock loci while down regulating the transcriptional activity of the majority of genes. Here, Sistonen and coworkers provide evidence of an elegant yet simple mechanism to coordinate both events that is the inhibition of the pause release of Pol II from nonstress promoters thereby creating a pool of free Pol II that is utilized at heat shock responsive promoters. Supporting the activities of heat shock genes are pre-assembled PICs (inferred based upon a previous study mapping the genomic positions of TBP (He et al. 2015) along with activated transcription factors such as HSF1. Intriguingly, the authors find that binding of HSF1, which does not recruit Pol II, correlated with SP2 occupancy suggesting SP2 might act as an insulator or directly block the ability of HSF1 nucleate Pol II. Overall, it is a well-done and exciting body of work.

1. Throughout the manuscript the authors make reference to the idea that they observe changes in the chromatin architecture that are helping to guide the heat shock transcriptional response. The abstract states “features of chromatin architecture that set the priming and control the activation..” yet little direct evidence is provide to substantiate this claim. While the authors do map where acetylated H4 is along the genome, this is just a posttranslational modification and might not represent an actual change to the chromatin architecture. Either the authors should provide a direct readout of chromatin architecture (minimally nucleosome presence/spacing could be addressed) or the text should be modified to simply refer to the H4 acetylation pattern. Statements such as “Besides uncovering the rapid opening of closed chromatin state upon stress...” should not be made unless direct evidence is provided.

In the manuscript, we have used the nucleotide resolution profiles of PRO-seq, GRO-cap and TBP ChIP-nexus to analyze specific features of promoter and enhancer architecture at divergent regulatory elements, primarily prior to stress. These analyses do not include histone occupancy or position

directly but they do show how the positioning of TBP (a key component of the Pre-Initiation Complex) and loading of RNA polymerase to a defined core promoter within a divergent regulatory element (measured with GRO-cap and PRO-seq) can establish directionality, support active transcription, and poise genes for rapid activation (measured with PRO-seq). To improve clarity, we modified the last sentence of the Abstract, now stating (lines 45-48): “These results reveal common and distinct features of chromatin architecture that orient transcription at divergent regulatory elements and prime transcriptional responses of promoters and enhancers genome-wide.”. We also added “prior to stress” into a sentence where the analyses of chromatin architecture are first described, and extended the sentence to define the features measured (lines 233-237 of the revised manuscript): “To address these questions, we mapped the architecture of divergent promoters prior to stress using the PRO-seq data of this study, together with GRO-cap (Core *et al.*, 2014) and ChIP-nexus of TATA-box binding protein (TBP) (He *et al.*, 2015), which together enable nucleotide-resolution profiling of the positioning, initiation, pausing, and elongation of Pol II.”. The nucleotide-resolution measures on TBP and RNA polymerase positions at divergent regulatory elements have been complemented with mapping the local enrichment of chromatin associated proteins, histone modifications and chromatin modifiers that are available in the ENCODE databases (Consortium EP, 2011, *PLoS Biol.* **9**: e1001046). The nucleotide-resolution maps of transcription-engaged RNA polymerases (prior to and upon stress) have enabled us to couple the promoter and enhancer architecture to the transcriptional response of the regulatory elements upon heat stress.

To monitor the change in the chromatin state upon heat stress, we have used histone H4 acetylation as a general read-out. Previously, acetylation of histone H4 has been mapped to open chromatin regions and proposed to loosen nucleosome-DNA interactions, allowing increased accessibility for transcription factors (reviewed e.g. by Galvani and Thiriet, 2015, *Genes [Basel]*. **6**: 607-621; Zentner and Henikoff, 2014, *Nat. Rev. Genet.* **15**: 814-827). Our results strongly support the previous reports that histone H4 acetylation occurs at accessible chromatin regions, as signal intensities from histone H4 acetylation ChIP-seq (our data) and DNaseI hypersensitivity (ENCODE data) positively correlate both along genes and at all histone H4 acetylated regions genome-wide (rho 0.72; see added lower panels of Supplementary Figure 3a of the revised manuscript. Please, see also below our reply to point 2 by Reviewer 1, where the comparability of the results obtained using the K562 cells of the ENCODE datasets to those using the K562 cells in our laboratories is addressed). We thank the Reviewer for raising this point. The new data in Supplementary Figure 3a and added discussion (lines 208-210) now provide the reader with an improved concept of the correlation of histone H4 acetylation and chromatin accessibility. We have, furthermore, taken great care not to directly claim chromatin openness based on the histone H4 acetylation, and followed the Reviewer’s advice by replacing “uncovering rapid opening of closed chromatin state” with “suggesting a rapid increase in chromatin accessibility” from the sentence mentioned above (lines 380-383 in the revised manuscript): “Besides suggesting a rapid increase in chromatin accessibility upon stress, the robust histone H4 acetylation at untranscribed regions demonstrated that histone-acetylation can be kinetically separated from, or occur regardless of, the transcriptional engagement of Pol II (**Fig. 7f**)”.

2. The authors often make use of the prior published genomic maps of the genomic binding patterns of different DNA binding proteins, which is a good and appropriate use of the public data. However, for the proteins they find critical to their model the authors should validate that these factors are

behaving in their cell line as expected. For example, by ChIP-PCR do the authors find the expected patterns of occupancy of TBP, CTCF, and SP2 at a few different sites?

This is a valid point, as cell lines may not be identical from one laboratory to another, and samples generated in different laboratories (or even within a single laboratory) could show differences in chromatin architecture and landscape that drive the transcriptional profile. The K562 cells used in this study display the phenotypic properties, including morphology and proliferation rate, that have been described in literature. Nevertheless, to rigorously assess the comparability of our results to those provided by the ENCODE laboratories, we globally analyzed the correlation of transcription profiles in K562 cells used in this study (PRO-seq data in non-heat shocked cells) to those cultured in the laboratories of Bernstein (Broad Institute), Snyder (Yale) and Iyer (UT Austin), utilizing the published Pol II ChIP-seq datasets from these three laboratories. Importantly, the transcriptional profile of K562 cells used in this study correlated extremely well (ρ 0.9) with the transcriptional profiles of K562 cells from the ENCODE laboratories (see new Supplementary Figure 6a and lines 309-321 of the revised manuscript), validating the use of K562 ENCODE datasets in our study. To specifically validate the applicability of CTCF, TBP and SP2 ChIP-seq datasets, we conducted additional analyses and experiments:

The ENCODE consortium has generated multiple ChIP-seq datasets for CTCF in K562 cells. Since ChIP-seq provides a global, high-resolution mapping of chromatin associated proteins, we utilized all the available CTCF ChIP-seq datasets to address the reproducibility of CTCF-binding at HSF1-targeted regions. Each of the ENCODE's CTCF ChIP-seq datasets reproduced the results shown in the manuscript. To illustrate this point, we added a new figure (Supplementary Figure 8a in the revised manuscript), showing a clear CTCF enrichment at HSF1-targeted promoters, dTREs and untranscribed sites with three additional ChIP-seq datasets (selected by their highest coverage). Furthermore, we included a browser shot image (Supplementary Figure 8b in the revised manuscript) depicting a strong CTCF-binding site (represented with three distinct datasets) at the gene body of an untranscribed gene (PLEKHG1) that recruits HSF1 upon stress. PLEKHG1 gene occurs, moreover, in a topologically associated domain (TAD; Rao *et al.*, 2015, *Cell* **159**: 1665-1680) with very little, if any, transcriptional activity as measured by PRO-seq in our K562 cells, further confirming the presence of CTCF at HSF1-targeted untranscribed regions. These results indicate that multiple laboratories have robustly, reproducibly and globally observed CTCF enrichment on a subset of sites where HSF1 is induced to bind upon heat shock. These results also confirm our conclusion of binding of a *trans*-activator to CTCF-enriched regions at untranscribed chromatin.

We have utilized two distinct datasets for TBP in K562 cells; TBP ChIP-seq (Consortium EP, 2011, *PLoS Biol.* **9**: e1001046) and TBP ChIP-nexus (Zeitlinger lab, He *et al.*, 2015, *Nat. Biotechnol.* **33**: 395-401). We added a new figure (Supplementary Figure 4d in the revised manuscript) where the signal intensities of TBP ChIP-seq and TPB ChIP-nexus are compared. Although ChIP-nexus clearly harbors superior resolution to ChIP-seq, also TPB ChIP-seq intensity strongly leans towards the sense strand when plotted against the mid-point of Pol II pause sites at divergent regulatory elements of highly transcribed and highly up-regulated genes. Supporting the polarized positioning of several PIC components at the divergent regulatory elements, average intensities of GTF2B and GTF2F1 ChIP-seq (Consortium EP, 2011, *PLoS Biol.* **9**: e1001046) also produced the positional preference of PIC components at the core promoter of the sense strand (see new Supplementary Figure 4e and lines 253-

259 in the revised manuscript). These results globally show the reproducibility of TBP positioning at the divergent regulatory elements, indicate that several components of PIC produce the binding pattern observed with TBP, and are supported by our PRO-seq data where high transcriptional directionality positively correlates with the polarized positioning of TBP, GTF2B and GTF2F1.

The ENCODE consortium has generated a single SP2 ChIP-seq dataset (in two replicates) in K562 cells. In order to validate that SP2 binds to expected regions in K562 cells cultured in our laboratories, we have performed ChIP-PCR using the same antibody (Santa Cruz sc-643) that was used to generate the ENCODE data (see Figure 1 for Reviewers only). For validation, we chose three distinct promoter regions: ARHGEF1 where SP2 has a high binding intensity (binding score 1000), HSPA1A where SP2 shows moderate binding (binding score 120), and ACTB (beta-actin) where no SP2 binding has been detected. In accordance to the results from the ENCODE consortium, we found that robust SP2 binding occurred at the promoters of ARHGEF1 and HSPA1A genes, whereas no binding of SP2 was detected at the promoter of beta-actin. Inducible HSF1-binding at the promoters of ARHGEF1 and HSPA1A is shown for reference, IgG was used as a negative antibody control and H3 as a positive antibody control.

3. The authors report on a potential role of SP2 as silencer of HSF1 transcriptional activity. Further characterization of the interaction between SP2 and HSF1 would considerably strengthen this point. For example, does inclusion of an SP2 site in an HSE-reporter lead to silence regardless of positioning or is it position dependent (e.g., does the SP2 site need to be between the HSE and start site)? Do SP2 and HSF1 physically interact?

The HSF1-bound promoters with high occupancy of SP2 provide intriguing genomic loci where the presence of the potent HSF1 *trans*-activator does not lead to transcriptional activation. We fully agree with the Reviewer that elucidating the mechanisms of a possible SP2-mediated regulation over HSF1 would require systematic analyses where the positioning of HSF1 and SP2 relative to one another, as well as to the PIC and paused Pol II, were determined. We are considering how to best approach this profound question, and envision that CRISPR/Cas9 experiments, where binding sites for SP2 or HSF1 are removed from or added to heat-responsive promoters, would rigorously address whether SP2 is responsible for the lack of *trans*-activation upon HSF1-binding. These analyses, together with other extensive experiments, are required to convincingly elucidate the mechanisms, and the possible positional constraints, of such a regulatory step in the context of the native chromatin. However, these experiments will likely take several months to be completed and they also extend beyond the scope of our present manuscript. In the revised manuscript, we modified the sentence suggesting SP2 as an insulator by replacing “suggests” with “opens intriguing possibilities to be addressed in future studies” (lines 330-334 of the revised manuscript): “The tight localization of SP2 between HSF1 and the complex containing paused Pol II and NELF-E (**Fig. 5e**) opens intriguing possibilities to be addressed in future studies on whether SP2 prevents HSF1 from gaining its *trans*-activator competence, or functions as a local insulator that restricts HSF1 from contacting the paused transcription machinery.”. Furthermore, in the Abstract, we removed the positional claim on SP2 to make space for a more detailed description of transcriptional directionality (see our answer to point 1

by Reviewer 2) and to tone down the positional relationship of SP2 with respect to HSF1 (see lines 44-45 in the revised manuscript): “Finally, HSF1-mediated trans-activation was restricted at promoters where Specificity Factor 2 (SP2) was abundant.”.

Reviewer #2

The authors have carried out an extensive study of the heat shock response in human cells and summarize the results and conclusions in a beautifully written manuscript. This is not just another global heat shock response paper. The analysis of genes, transcribed enhancers, and non-transcribed enhancers by PRO-Seq, CHIP-Seq is for the most part very clear and interesting. This is the authors description of their findings:

- 1) Cells elicit a massive response to stress by reprogramming transcription and adjusting the chromatin landscape of genes and enhancers.*
- 2) A gene's response to stress is defined at the pause release of promoter-proximal Pol II.*
- 3) Strand-specific assembly of PIC and Pol II sets up the orientation of promoters and pre-wires transcriptional activation.*
- 4) Lineage-specific factors prime enhancer transcription.*
- 5) Potent trans-activator HSF1 localizes to transcription-primed and architectural protein-primed genomic regions.*
- 6) Local SP2-associated chromatin architecture can restrain the activation of primed, HSF1-targeted gene promoters.*

All of these points are important and well supported by the data.

Specific comments:

- 1. In the abstract the term “of the coding strand” is used. I think the authors mean “for the sense direction”*

We removed “of the coding strand” and further modified the sentence. Moreover, we now introduce the concept of transcriptional regulation from divergent regulatory elements already in the Abstract (see lines 38-41 in the revised manuscript): “Divergent transcription from promoters and enhancers is

known to be generated by a pair of tightly-linked core initiation regions. We found that highly-transcribed and highly-inducible genes display both selective assembly of general transcription factors on the core sense promoters, and strong transcriptional directionality.”.

2. I didn't understand the graph of TBP in Fig. 4b. Why is TBP downstream of the paused Pol II in the sense direction. Sign seems backwards.

We thank the Reviewer for pointing out the lack of clarity in the illustration in Figure 4b. We do see how the arrows can be misinterpreted. At each divergent regulatory element, we have mapped the Pol II pause site at the sense and anti-sense strands. Zero in the x-axis of Figure 4b indicates the mid-nucleotide between these pause sites, allowing for a fair comparison between the signal intensities of the core promoters on both strands. To better illustrate the position where signal intensity has been mapped, we added a schematic figure (right panel of Figure 4a in the revised manuscript), depicting the Pol II signal at the sense and anti-sense strands, and indicating the transcription initiation sites, Pol II pause sites, as well as the mid-coordinate between the pause sites. For consistency and clarity, we modified the arrows on top of the metaprofiles or heatmaps in Figure 4b,c and Supplementary Figure 4a,c-e.

Overall, I really enjoyed reading this manuscript. I want my friends to read it too!

Figure 1 for Reviewers only. Validation of SP2-binding using ChIP-PCR. **a)** Table showing the binding scores (relative number, 0-1000) of HSF1 and SP2 to the promoters of HSPA1A, ARHGEF1 and ACTB (beta-actin). Response denotes the transcriptional response of the gene to a 30-minute heat shock at 42°C. Down: down-regulated; Up: up-regulated. **b)** Browser shot images of ChIP-seq data for HSF1 (NHS and HS30; our study), and SP2 (NHS; ENCODE) at the promoters of HSPA1A, ARHGEF1 and ACTB. The y-scale is indicated at the upper right corner of each track line. NHS: non-heat shock condition; HS30: 30 min at 42°C. **c)** ChIP-PCR of HSF1 and SP2 at the promoters of HSPA1A, ARHGEF1 and ACTB. The antibodies (Enzo ADI-SPA 901 for HSF1; Santa Cruz sc-643 for SP2) that were used for generating the ChIP-seq datasets were also used in ChIP-PCR validation. IgG (Santa Cruz sc-2027) was used as a negative antibody control and histone H3 (AbCam, ab1791)

as a positive antibody control. Input is the whole fragmented genome. The primers for HSPA1A and ACTB promoters have been reported earlier (Östling *et al.*, 2007, *J. Biol. Chem.* **282**: 7077-7086; Vihervaara *et al.* 2013, *P.N.A.S.*, **110**: E3388-3397), the primers for the ARHGEF1 promoter have sequences 5'-gcgggaggagtagagtcgt-3' and 5'-ccctgagtcacacattgc-3'.

REVIEWERS' COMMENTS:

Reviewer #1 (Remarks to the Author):

The authors have sufficiently responded to the concerns raised in the initial review. The manuscript now stands as an important contribution to the fields of stress signaling and transcription.